# Sex peptide targets distinct higher order processing neurons in the brain to induce the female post-mating response

Mohanakarthik P Nallasivan[1], Deepanshu ND Singh[2], Mohammed Syahir RS Sahir[1], Matthias Soller[1,2]*

[1]School of Biosciences, College of Life and Environmental Sciences, University of Birmingham, Birmingham, United Kingdom; [2]Division of Molecular and Cellular Function, School of Biological Sciences, University of Manchester, Manchester, United Kingdom

## eLife Assessment

This study delivers **valuable** new insights into the neural circuits involved in post-mating responses (PMR) in *Drosophila* females, supported by **convincing** evidence that the circuits for mating receptivity and egg laying are distinct. The new experimental evidence adds to the current understanding of the neural circuits and molecular mechanisms underpinning PMR.

*For correspondence:
matthias.soller@manchester.ac.uk

Competing interest: The authors declare that no competing interests exist.

**Abstract** Sex peptide (SP) transferred during mating induces female post-mating responses including refractoriness to re-mate and increased oviposition in *Drosophila*. Yet, where SP-target neurons reside remained uncertain. Here, we show that expression of membrane-tethered SP (mSP) predominantly in the head or trunk either reduces receptivity or increases oviposition, respectively. Using fragments from large regulatory regions of *Sex Peptide Receptor*, *fruitless*, and *doublesex* genes together with intersectional expression of mSP, we identified distinct interneurons in the brain and abdominal ganglion controlling receptivity and oviposition. These SP response-inducing neurons (SPRINz) can induce post-mating responses through SP received by mating. Trans-synaptic mapping of neuronal connections reveals input from sensory processing neurons and two post-synaptic trajectories as output. Hence, SP-target neurons operate as key integrators of sensory information for decision-making of behavioural outputs. Multi-modularity of SP-targets further allows females to adjust SP-mediated male manipulation to physiological state and environmental conditions for maximising reproductive success.

## Introduction

Reproductive behaviours are to a large degree hard-wired in the brain to guarantee reproductive success, making the underlying neuronal circuits amenable to genetic analysis (*Dulac and Kimchi, 2007*; *Yamamoto and Koganezawa, 2013*; *Anderson, 2016*; *Rings and Goodwin, 2019*).

During development, sex-specific circuits are built into the brain under the control of the sex determination genes *doublesex* (*dsx*) and *fruitless* (*fru*) in *Drosophila* (*Schütt and Nöthiger, 2000*; *Billeter et al., 2006*). They encode transcription factors that are alternatively spliced in a male or female-specific mode (*Schütt and Nöthiger, 2000*). By default, the *dsx* gene generates the male-specific isoform Dsx$^M$, while a female-specific isoform Dsx$^F$ is generated by alternative splicing and expressed in about ~700 distinct neurons in the brain important for female reproductive behaviours directing readiness to mate and egg laying (*Rideout et al., 2010*; *Rezával et al., 2012*). Fru$^M$ is expressed in

about ~1000 neurons in males and implements development of neuronal circuitry key to display male courtship behaviour, but is switched off in females through alternative splicing by incorporation of a premature stop codon (*Demir and Dickson, 2005*; *Manoli et al., 2005*; *Stockinger et al., 2005*).

The circuitry of female-specific behaviours, including receptivity to courting males for mating and egg laying, has been mapped using intersectional gene expression via the *split-GAL4* system to restrict expression of activators or inhibitors of neuronal activity to very few neurons (*Aranha and Vasconcelos, 2018*; *Wang et al., 2020a*; *Wang et al., 2020b*; *Wang et al., 2021*; *Cury and Axel, 2023*). Through this approach, sensory neurons in the genital tract have been identified as key signal transducers for the readiness to mate and the inhibition of egg laying connecting to central parts of the brain via projection to abdominal ganglion neurons (*Häsemeyer et al., 2009*; *Yang et al., 2009*; *Rezával et al., 2012*; *Feng et al., 2014*). This circuit then projects onto centrally localised pattern generators in the brain to direct a behavioural response via efferent neurons (*Wang et al., 2020a*; *Wang et al., 2020b*; *Wang et al., 2021*).

Once females have mated, they will reject courting males and lay eggs (*Manning, 1967*). Post-mating responses (PMRs) are induced by male-derived sex peptide (SP) and other substances trans-ferred during mating (*Chen et al., 1988*; *Avila et al., 2011*; *Hopkins and Perry, 2022*; *Kim et al., 2024*; *Singh and Soller, 2025*). In addition to refractoriness to remate and oviposition, SP will induce a number of other behavioural and physiological changes, including increased egg production, feeding, a change in food choice, sleep, memory, constipation, midgut morphology, stimulation of the immune system, and sperm storage and release (*Soller et al., 1999*; *Peng et al., 2005*; *Carvalho et al., 2006*; *Domanitskaya et al., 2007*; *Kim et al., 2010*; *Ribeiro and Dickson, 2010*; *Scheunemann et al., 2019*; *Cognigni et al., 2011*; *Avila et al., 2010*; *Isaac et al., 2010*; *Wainwright et al., 2021*; *White et al., 2021*). SP binds to broadly expressed sex peptide receptor (SPR), an ancestral receptor for myoinhibitory peptides (MIPs) (*Yapici et al., 2008*; *Kim et al., 2010*; *Jang et al., 2017*). Although MIPs seem not to induce PMRs, excitatory activity of MIP-expressing neurons underlies re-mating (*Yapici et al., 2008*; *Kim et al., 2010*; *Jang et al., 2017*). Expression of membrane-tethered SP (mSP) induces PMRs in an autocrine fashion when expressed in neurons, but not glia (*Nakayama et al., 1997*; *Haussmann et al., 2013*).

First attempts to identify SP target neurons by enhancer *GAL4*-induced expression of *UASmSP* only identified lines with broad expression in the nervous system (*Nakayama et al., 1997*). Later, drivers with more restricted expression, including *dsx*, *fru*, and *pickpocket* (*ppk*) genes were identi-fied, but they are expressed in all parts of the nervous system throughout the body, eluding to reveal the location of SP target sites unambiguously (*Yapici et al., 2008*; *Häsemeyer et al., 2009*; *Yang et al., 2009*; *Rezával et al., 2012*; *Haussmann et al., 2013*).

To delineate where in the *Drosophila* SP target neurons are located which induce the main PMRs, refusal to mate and egg laying, we expressed mSP predominantly in the head or trunk. These exper-iments separate reduction of receptivity induced in the head from trunk induction of egg laying. To further restrict our search for SP target neurons, we focused on three genes, *SPR*, *dsx*, and *fru*, because SPR is broadly expressed but anticipated to induce PMRs only from few neurons, and because *GAL4* inserted in the endogenous *dsx* and *fru* loci induces PMRs from mSP expression. Using *GAL4* tiling lines with fragments encompassing the regulatory regions of complex *SPR*, *fru*, and *dsx* genes (*Pfeiffer et al., 2008*; *Jenett et al., 2012*; *Kvon et al., 2014*), we identified one regulatory region in each gene reducing receptivity and inducing egg laying upon mSP expression, and one additional region in *SPR* only inducing egg laying. To further refine this analysis, we used intersectional gene expression using *split-GAL4* and *flipase* (*flp*)-mediated excision of stop cassettes in *UAS* reporters (*Struhl and Basler, 1993*; *Luan et al., 2006*). Consistent with previous results that the SP response can be induced via multiple pathways (*Haussmann et al., 2013*), we found distinct sets of SP response-inducing neurons (SPRINz) in the central brain and the abdominal ganglion that can induce PMRs via expression of mSP either reducing receptivity and inducing egg laying, or affecting only one of these PMRs. In contrast, we identified genital tract neuron expressing lines including *split-GAL4 nSyb ∩ ppk* that did not induce PMRs by expression of mSP. Likewise, we find expression of mSP or neuronal acti-vation in head sex peptide sensing neurons (SPSN) neurons can induce PMRs. Mapping the pre- and post-synaptic connections of the distinct SP target neurons by *retro-* and *trans*-Tango (*Talay et al., 2017*; *Sorkaç et al., 2023*) revealed that SP target neurons direct higher order sensory processing in the central brain. These neurons feed into two common post-synaptic neuronal subtypes indicating

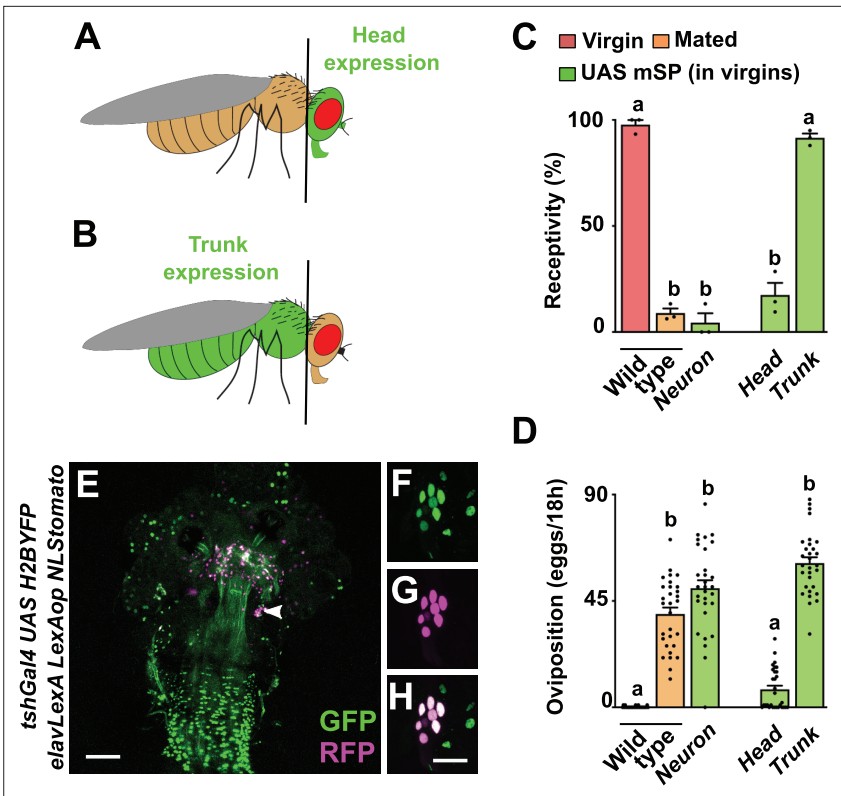

**Figure 1.** The main post-mating responses (PMRs) in females can be separated. (**A, B**) Schematic depiction of head and trunk expression in *Drosophila elav FRTstopFRT GAL4; otdflp* (**A**) and in *tshGAL4* (**B**) visualised by *UAS GFP* (green). (**C, D**) Receptivity (**C**) and oviposition (**D**) of wild type control virgin (red) and mated (orange) females, and virgin females expressing *UAS mSP* (green) pan-neuronally with *nsybGAL4* or in head and trunk patterns shown as means with standard error from three repeats for receptivity (21 females per repeat) by counting the number of females mating within a 1-hour period or for oviposition by counting the eggs laid within 18 hours from 30 females. Statistically significant differences from ANOVA post hoc comparison are indicated by different letters (p<0.0001). (**E–H**) Representative adult female genital tract showing *tshGAL4 UAS H2BYFP* (green) and *elavLexA LexAop NLStomato* (red) nuclear expression. The magnification (**F–H**) shows sensory genital tract neurons. Scale bars shown in (**E**) and (**H**) are 100 μm and 20 μm, respectively.

The online version of this article includes the following source data and figure supplement(s) for figure 1:

**Source data 1.** Quantitative results used to generate graphs in *Figure 1C and D*.

**Figure supplement 1.** Analysis of head and trunk expression lines.

that SP interferes with the integration of diverse sensory inputs to build a stereotyped output either reducing receptivity and/or increasing egg laying.

## Results

### Reduction of receptivity and induction of egg laying are separable by head and trunk expression of membrane-tethered SP

Due to the complex behavioural and physiological changes induced by SP, neurons in the central nervous system have been suspected as main targets for SP (*Kubli, 1992*). To express mSP only in the head, we used an *elav FRTstopFRT GAL4* in combination with *otdflp* that expresses in the head to drive recombination and head-specific expression of mSP from *UAS* (*Figure 1A*, *Figure 1—figure supplement 1A–F*; *Haussmann et al., 2008*; *Asahina et al., 2014*; *Zaharieva et al., 2015*; *Nallasivan et al., 2021*). To express mSP predominantly in the trunk, we used *tshGAL4* (*Figure 1B*, *Figure 1— figure supplement 1G–L*; *Soller et al., 2006*).

When we expressed mSP in the head, females reduced receptivity indistinguishable from mated females, but did not lay eggs, thereby again demonstrating that the two main PMRs can be separated (*Figure 1C and D*; *Haussmann et al., 2013*). In contrast, when we expressed mSP in the trunk, females remained receptive but laid eggs in numbers indistinguishable from mated females (*Figure 1C and D*).

Moreover, *tshGAL4* is expressed in *fru*, *dsx*, *ppk* genital tract sensory neurons (*Figure 1E–H*). Since mSP expression with *tshGAL4* does not affect receptivity, these genital tract neurons unlikely are direct targets for SP (*Haussmann et al., 2013*). Taken together, these results indicate the presence of SP target neurons in the brain and ventral nerve cord (VNC) for the reduction of receptivity and induction of egg laying, respectively.

## Few restricted regulatory regions in large *SPR*, *fru,* and *dsx* genes can induce the SP response

Expression of mSP from *UAS* via *GAL4* inserts in *fru* and *dsx* genes induces a robust reduction in receptivity and increase in egg laying (*Rezával et al., 2012*; *Haussmann et al., 2013*). To identify SP target neurons, we thought to dissect the broad expression pattern of complex *SPR*, *fru*, and *dsx* genes spanning 50–80 kb by identifying regulatory DNA fragments in the enhancer regions that drive *UAS mSP* in a subset of neurons. For these experiments, we analysed 22, 27, and 25 *GAL4* lines from the VDRC and Janelia tiling *GAL4* projects (*Pfeiffer et al., 2008*; *Jenett et al., 2012*; *Kvon et al., 2014*; *Figure 2A–C*).

Strikingly, in *SPR*, *fru*, and *dsx* genes, we identified only one regulatory region in each gene (*SPR8*, *fru11/12*, and *dsx24*) that reduced receptivity and induced egg laying through *GAL4 UAS* expression of mSP (*Figure 2D and E*). In addition, we identified one line (*SPR12*) in the *SPR* gene that induced egg laying but did not reduce receptivity, consistent with previous results that SP regulation of receptivity and egg laying can be split (*Haussmann et al., 2013*).

All of these lines expressed in subsets of neurons in the central brain and the VNC in distinct, but reduced patterns compared to the expression of the *SPR*, *fru*, and *dsx* genes (*Yapici et al., 2008*; *Rideout et al., 2010*; *Zhou et al., 2014*; *Figure 2F–O*). Moreover, these lines showed prominent labelling of abdominal ganglion neurons in the VNC (*Figure 2K–O*). In addition, all of these lines except *SPR12* are also expressed in genital tract sensory neurons (*Figure 2—figure supplement 1A–E*).

From all the 74 lines that we have analysed for PMRs from *SPR*, *fru,* and *dsx* genes, we also analysed expression in genital tract sensory neurons as they had been postulated to be the primary targets of SP (*Yapici et al., 2008*; *Häsemeyer et al., 2009*; *Yang et al., 2009*; *Rezával et al., 2012*). Apart from PMR-inducing lines *SPR8*, *fru11*, *fru12,* and *dsx24*, that showed expression in genital tract sensory neurons, we identified three lines (*SPR3*, *SPR 21*, and *fru9*), which also robustly expressed in genital tract sensory neurons but did not induce PMRs from expression of mSP (*Figure 3*, *Figure 2—figure supplement 2*).

## Genital tract neurons do not mediate changes in receptivity and oviposition by mSP

Since genital tract sensory neurons have been postulated to induce the SP response, we tested previously identified *split-GAL4* (SPSN-1: VT058873 ∩ VT003280/FD6 and SPSN-2: VT58873 ∩ VT033490) lines, which upon neuronal inhibition reduced receptivity and induced egg laying (*Feng et al., 2014*), for their capacity to induce the SP response upon expression of mSP. Both lines reduced receptivity and induced egg laying upon expression of mSP (*Figure 3A and B*). However, since genital tract neuron expressing *SPR3* and *SPR21* lines did not induce PMRs upon expression of mSP, the SP response induced by *SPSN1* and *2 split-GAL4 mSP* expression could originate from other neurons.

Expression analysis of these two lines revealed that in addition to expression in genital tract sensory neurons (*Figure 3C, D, G, and H*), they also showed expression in the brain and VNC (*Figure 3K, L, O, and P*). Intriguingly, the brain neurons labelled in SPSN-1 resembled the neurons identified by *SPR8* ∩ *FD6* (*Figure 4G*).

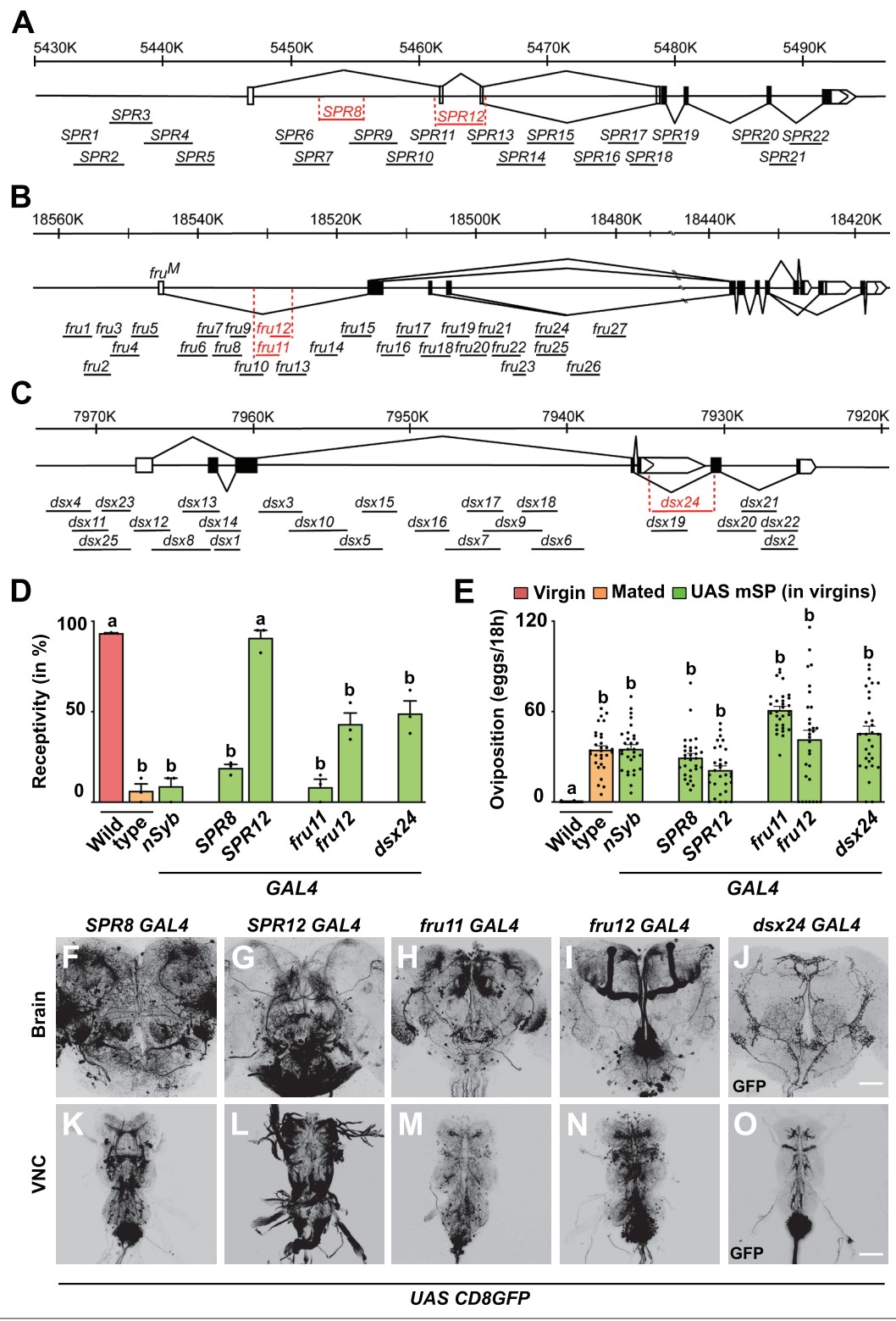

**Figure 2.** Distinct regulatory regions in *SPR*, *fru,* and *dsx* genes induce post-mating responses (PMRs) from mSP expression. (**A–C**) Schematic representation of *SPR*, *fru,* and *dsx* chromosomal regions depicting coding and non-coding exons as black or white boxes, respectively, and splicing patterns in solid lines. Vertical lines below the gene model depict enhancer *GAL4* lines with names and those in red showed PMRs by expression of mSP. (**D, E**) Receptivity (**D**) and oviposition (**E**) of wild type control virgin (red) and mated (orange) females, and virgin females

*Figure 2 continued on next page*

*Figure 2 continued*

expressing *UAS mSP* (green) under the control of GAL4 pan-neuronally in *nsyb* or in *SPR8, SPR12, fru11, fru12*, and *dsx24* patterns shown as means with standard error from three repeats for receptivity (21 females per repeat) by counting the number of females mating within a 1-hour period or for oviposition by counting the eggs laid within 18 hours from 30 females. Statistically significant differences from ANOVA post hoc comparison are indicated by different letters (p≤0.0001). (**F–O**) Representative adult female brains (**F–J**) and ventral nerve cords (VNC, **K–O**) expressing *UAS CD8GFP* under the control of *SPR8, SPR12, fru11, fru12,* and *dsx24 GAL4*. Scale bars shown in (**J**) and (**O**) are 50 μm and 100 μm, respectively.

The online version of this article includes the following source data and figure supplement(s) for figure 2:

**Source data 1.** Quantitative results used to generate graphs in *Figure 2D and E*.

**Figure supplement 1.** Expression analysis of PMR-inducing *GAL4* in the genital tract.

**Figure supplement 2.** Expression analysis of non-PMR-inducing *fru9GAL4* in the genital tract.

## Secondary ascending abdominal ganglion neurons can induce the PMRs from mSP expression

A screen aiming to identify neurons involved in the control of receptivity and egg laying by expression of the rectifying potassium channel Kir2.1 identified six enhancer *GAL4* driver lines (*FD1-6*) (*Feng et al., 2014*). *FD1-6* are expressed in diverse subsets of neurons in the brain and the VNC; in particular, they show common expression in the abdominal ganglion with projections to the central brain. The lines expressing in FD1-5 neurons have been termed SAG (secondary ascending abdominal ganglion neurons) neurons that are also interconnected with MIP sensing neurons (*Jang et al., 2017*). Since enhancer lines identified in *SPR*, *fru*, and *dsx* genes are prominently expressed in the abdominal ganglion, we tested whether mSP expression from these FD1-6 lines induced PMRs.

From these six lines, one robustly suppressed receptivity and induced egg laying (*FD6/VT003280*), while two lines only induced egg laying (*FD3/VT4515* and *FD4/V000454*) similar to controls from mSP expression (*Figure 4*). Again, all three lines also expressed in subsets of neurons in the central brain and VNC, particularly in the abdominal ganglion (*Zhou et al., 2014*). In addition, *FD3* and *FD4* did not express in genital tract sensory neurons, in contrast to *FD6* (*Feng et al., 2014*). A SAG split-GAL4 (*VT050405/FD1 AD* and *VT007068/FD2 DBD*) line did not show a response to expression of mSP and virgin females, for example, they mated and did not lay eggs (*Figure 4*).

## Intersectional expression reveals distinct mSP-responsive neurons in the central brain and abdominal ganglion

To further restrict the expression to fewer neurons, we intersected the expression patterns of those lines that induced robust reduction of receptivity and increase of egg laying using *split-GAL4* (*SPR8*, *fru11/12*, *dsx*, and *FD6*; for further experiments we used *dsxGAL4-DBD*, because *dsx24* is less robust and *fru11* and *fru12* were made into one fragment) that activates the *UAS* reporter when *GAL4* is reconstituted via dimerisation of activation (AD-GAL4) and DNA binding (GAL4-DBD) domains (*Luan et al., 2006*; *Figure 5A*).

Again, the intersection of *SPR8* with *fru11/12*, *dsx* or *FD6*, and *fru11/12* with *dsx* or *FD6* expression robustly reduced receptivity and increased egg laying upon expression of mSP (*Figure 5B and C*). Accordingly, we termed these SP response-inducing neurons SPRINz, though the exact identity in the *split-GAL4* intersection population needs to be determined.

When we further analysed the expression of these *split-GAL4* intersections in the brain, we found that each combination first showed very restricted expression, but second, that none of these combinations labelled the same neurons (*Figure 5D–H*). For *dsx* neurons, *split-GAL4* intersections correspond to a subset of dPC2l (*SPR8* ∩ *dsx*) and dPCd-2 (*fru11/12* ∩ *dsx*) neurons (*Deutsch et al., 2020*; *Schretter et al., 2020*; *Nojima et al., 2021*). These results suggest the SP targets interneurons in the brain that feed into higher processing centres from different entry points likely representing different sensory input.

In the VNC, we found expression in the abdominal ganglion with all *split-GAL4* combinations (*Figure 5I–M*). In particular, the intersection of *dsx* with *SPR8* or *fru11/12* showed exclusive expression in the abdominal ganglion, while the other combinations also expressed in other cells of the VNC.

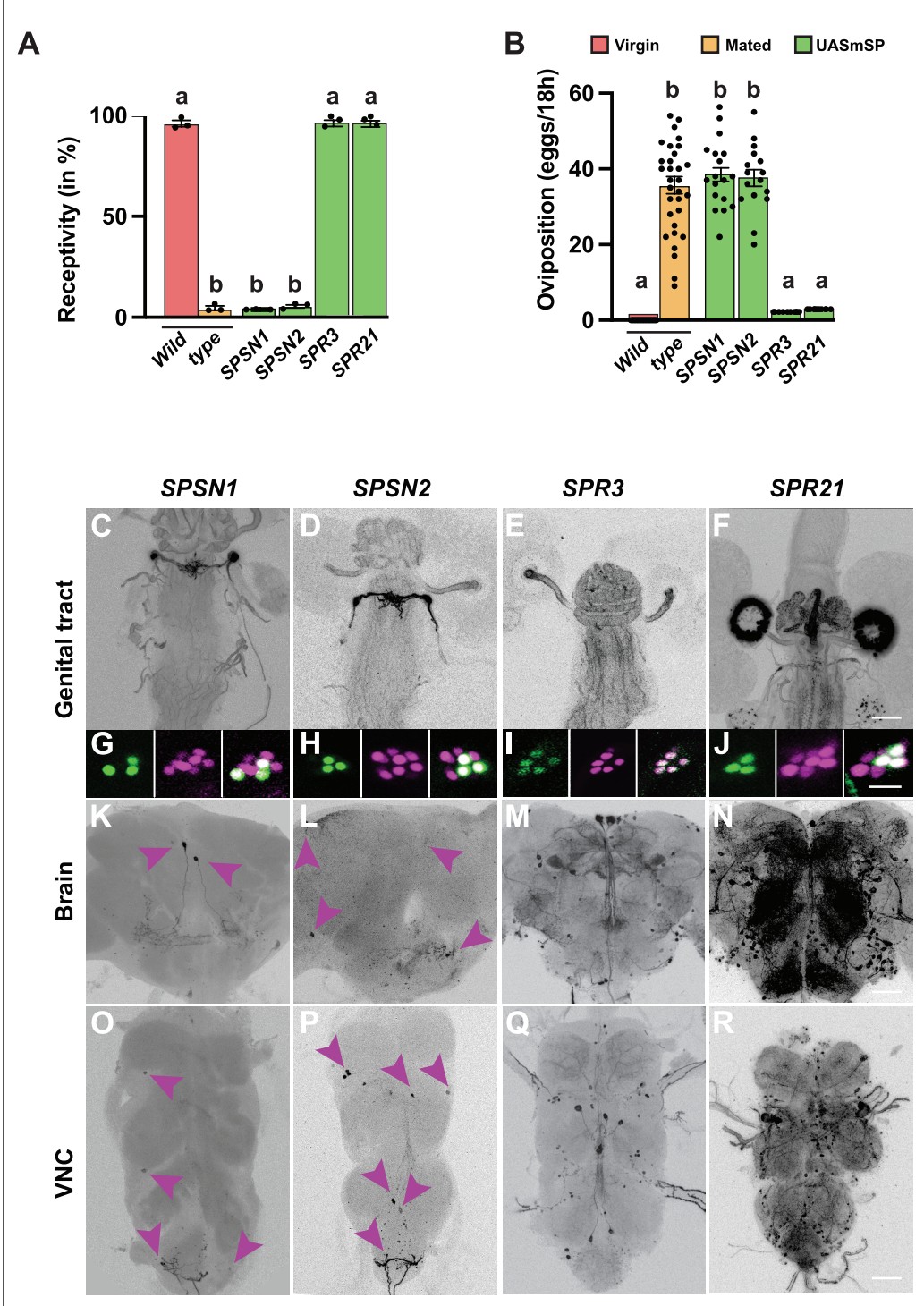

**Figure 3.** Expression of mSP in *SPSN* and genital tract expressing *SPR* lines does not support a major role for genital tract neurons in inducing the sex peptide response. (**A, B**) Receptivity (**A**) and oviposition (**B**) of wild type control virgin (red) and mated (orange) females, and virgin females expressing *UAS mSP* (green) under the control of *SPSN 1* and *SPSN2*, and *SPR3* and *SPR9 GAL4* lines shown as means with standard error from three repeats for receptivity (21 females per repeat) by counting the number of females mating within a 1-hour period or for oviposition by counting the eggs laid within 18 hours from 30 females. Statistically significant differences from ANOVA post hoc comparison are indicated by different letters (p<0.0001). (**C–J**) Representative genital tracts labelled with UAS CD8 GFP and genital tract neurons labelled with UAS H2BYFP and *elavLexA AopNLStomato*.

*Figure 3 continued on next page*

*Figure 3 continued*

(**K–R**) Adult female brains (**K–N**) and ventral nerve cords (VNC. **O–R**) expressing *UAS CD8GFP*. Scale bars shown in (**F, J, N, R**) are 100 μm, 20 μm, 50 μm and 100 μm, respectively.

The online version of this article includes the following source data for figure 3:

**Source data 1.** Quantitative results used to generate graphs in *Figure 3A and B*.

Altogether, these data suggest that the abdominal ganglion harbours several distinct types of neurons involved in directing PMRs (*Oliveira-Ferreira et al., 2023*).

In the female genital tract, these *split-Gal4* combinations show expression in genital tract neurons with innervations running along oviduct and uterine walls (*Figure 5—figure supplement 1A–J*). In addition, *SPR8 ∩ fru11/12* and *SPR8 ∩ dsx* were also expressed in the spermathecae (*Figure 5—figure supplement 1A and B*).

## mSP-responsive neurons rely on SPR and are required for PMRs induced by SP delivered through mating

Next, we tested whether PMRs induced by mSP expression in the *SPR8 ∩ dsx, fru11/12 ∩ dsx* or *SPR8 ∩ fru11/12* rely on *SPR*. Expression of mSP in *dsx ∩ SPR8* and *dsx ∩ fru11/12* neurons in *SPR* mutant females did not reduce receptivity or induce egg laying (*Figure 6A and B*, see also *Figure 5A and B*), while a partial response was observed for *SPR8 ∩ fru 11/12* induced mSP expression in *SPR* mutant females, which is consistent with presence of additional receptors for SP (*Haussmann et al., 2013*).

Since SP is transferred during mating to females and enters the hemolymph (*Haussmann et al., 2013*), we wanted to test whether SPR is required in these neurons for inducing PMRs after mating. For *SPR RNAi* in *dsx ∩ fru11/12* and *SPR8 ∩ fru 11/12* neurons, no reduction, or a partial reduction, of receptivity was observed, respectively, while *SPR RNAi* in *dsx ∩ SPR8* neurons turned virgin females unreceptive (*Figure 6A*). Expression of mSP in *dsx ∩ fru11/12* neurons in the context of *SPR RNAi* partially reduced receptivity, again suggesting additional receptors for SP (*Haussmann et al., 2013*).

Strikingly, however, *SPR RNAi* in these neurons prevented egg laying independent of whether SP was delivered by mating or when tethered to the membrane of these neurons (*Figure 6B*).

These results demonstrate that neurons identified by *split-GAL4* intersected expression of *SPR8* with *dsx* or *fru11/12*, or *fru11/12* with *dsx* are genuine SP targets as they rely on *SPR* and PMRs are induced by SP delivered through mating.

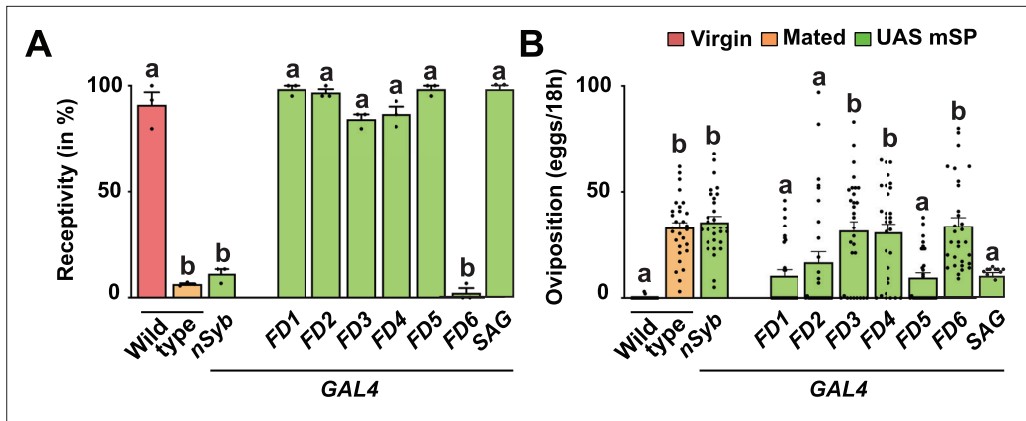

**Figure 4.** Expression of membrane-tethered sex peptide (mSP) in secondary ascending abdominal ganglion neurons induces post-mating responses (PMRs). (**A, B**) Receptivity (**A**) and oviposition (**B**) of wild type control virgin (red) and mated (orange) females, and virgin females expressing *UAS mSP* (green) under the control of GAL4 pan-neuronally in *nsyb* or in *FD1, FD2, FD3, FD4, FD5,* and *FD6,* or with *SAG split-Gal4* patterns shown as means with standard error from three repeats for receptivity (21 females per repeat) by counting the number of females mating within a 1-hour period or for oviposition by counting the eggs laid within 18 hours from 30 females. Statistically significant differences from ANOVA post hoc comparison are indicated by different letters (p<0.0001).

The online version of this article includes the following source data for figure 4:

**Source data 1.** Quantitative results used to generate graphs in *Figure 4A and B*.

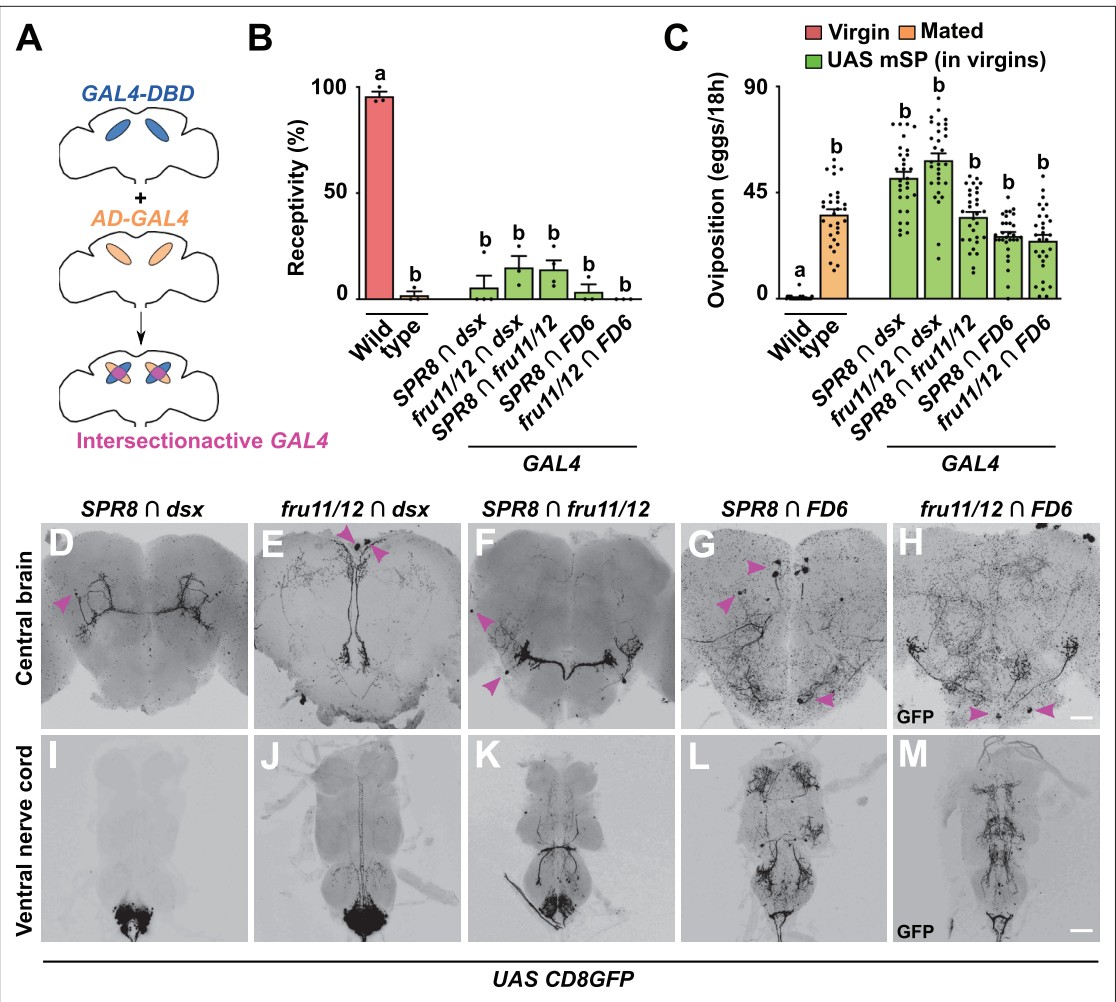

**Figure 5.** Distinct circuits from the intersection of *SPR*, *fru*, *dsx*, and *FD6* patterns in the brain and ventral nerve cord (VNC) induce post-mating responses (PMRs) from membrane-tethered sex peptide (mSP) expression. (**A**) Schematic showing the intersectional gene expression approach: GAL4 activation (AD, orange) and DNA binding domains (DBD, blue) are expressed in different, but overlapping patterns. Leucine zipper dimerisation reconstitutes a functional *split-GAL4* in the intersection (pink) to express *UAS* reporters. (**B, C**) Receptivity (**B**) and oviposition (**C**) of wild type control virgin (red) and mated (orange) females, and virgin females expressing *UAS mSP* (green) under the control of *split-GAL4* intersecting *SPR8 ∩ fru11/12*, *SPR8 ∩ dsx*, *SPR8 ∩ FD6*, *fru11/12 ∩ dsx*, and *fru11/12 ∩ FD6* patterns shown as means with standard error from three repeats for receptivity (21 females per repeat) by counting the number of females mating within a 1-hour period or for oviposition by counting the eggs laid within 18 hours from 30 females. Statistically significant differences from ANOVA post hoc comparison are indicated by different letters (p<0.0001). (**D–M**) Representative adult female brains and VNC expressing *UAS CD8GFP* under the control of *SPR8 ∩ fru11/12*, *SPR8 ∩ dsx*, *SPR8 ∩ FD6*, *fru11/12 ∩ dsx*, and *fru11/12 ∩ FD6*. Scale bars shown in (**H**) and (**M**) are 50 µm and 100 µm, respectively.

The online version of this article includes the following source data and figure supplement(s) for figure 5:

**Source data 1.** Quantitative results used to generate graphs in *Figure 5B and C*.

**Figure supplement 1.** Expression analysis of *split-GAL4* in the genital tract.

## Expression of mSP in distinct neurons in the brain induces PMRs

The analysis of *ppkGAL4* neurons in SP-insensitive *Nup54* alleles revealed a hierarchy of trunk neurons that dominate over central brain neurons (*Nallasivan et al., 2021*). To focus on the role of central brain neurons, we generated a *UAS mSP* line with a stop cassette (*UAS FRTstopFRT mSP*) that allows us to restrict expression of mSP to the head in the presence of *otdflp*, which only expresses in the head (*Figure 7A*), but not in the trunk (*Asahina et al., 2014*; *Nallasivan et al., 2021*).

In combination with the intersectional approach, we now can restrict mSP expression to few central brain neurons, or alternatively activate or silence these neurons (*Figure 7B*). Expression of mSP in *SPR8 ∩ dsx*, *fru11/12 ∩ dsx*, or *SPR8 ∩ fru11/12* neurons in the central brain significantly reduced

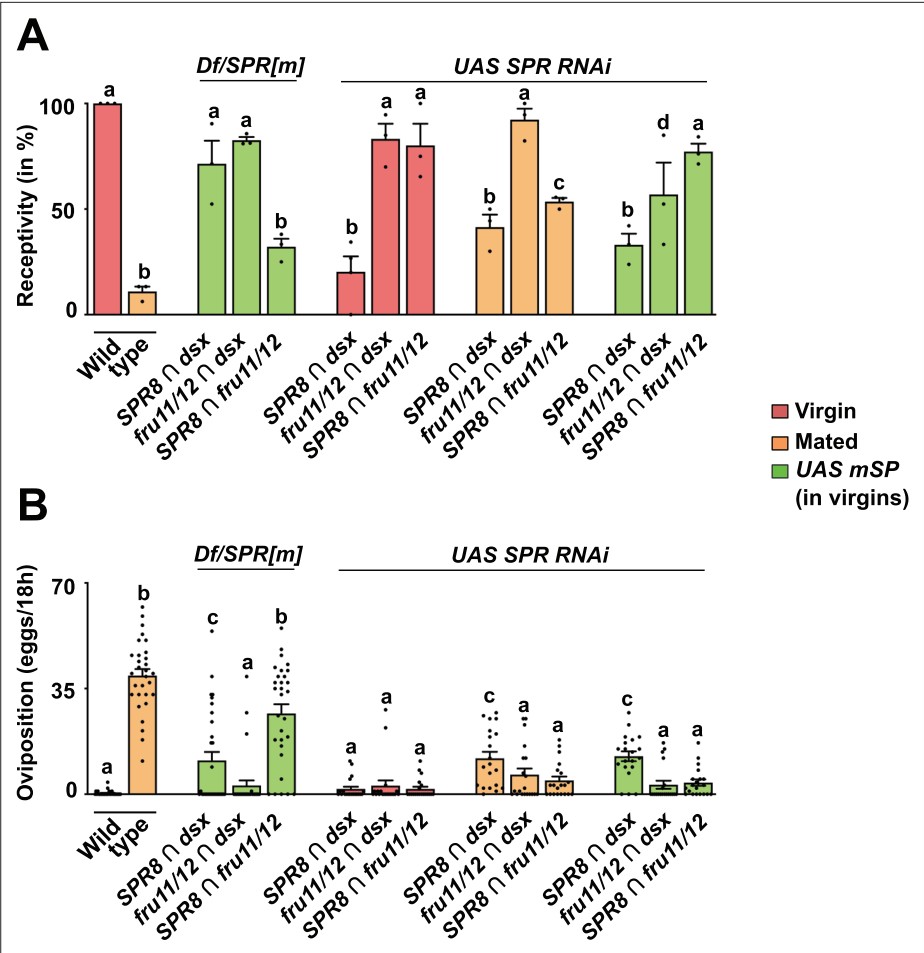

**Figure 6.** Distinct neuronal circuitries from the intersection of *SPR*, *fru*, and *dsx* sense sex peptide (SP) after mating to induce post-mating responses (PMRs). (**A, B**) Receptivity (**A**) and oviposition (**B**) of wild type control virgin (red) and mated (orange) females, and virgin females expressing *UAS mSP* (green) under the control of *split-Gal4* intersecting *SPR8 ∩ dsx*, *fru11/12 ∩ dsx*, and *SPR8 ∩ fru11/12* patterns in *SPR/Df* mutant females or *SPR* RNAi knock-down shown as means with standard error from three repeats for receptivity (21 females per repeat) by counting the number of females mating within a 1-hour period or for oviposition by counting the eggs laid within 18 hours from 30 females. Statistically significant differences from ANOVA post hoc comparison are indicated by different letters (p<0.0001 except p=0.002 and p=0.006 for c and d in **A**, and *P*=0.004 for c in **B**).

The online version of this article includes the following source data for figure 6:

**Source data 1.** Quantitative results used to generate graphs in *Figure 6A and B*.

receptivity, but oviposition was only substantially induced in *SPR8 ∩ dsx* brain neurons (*Figure 7C and D*). In *fru11/12 ∩ dsx* or *SPR8 ∩ fru11/12*, PMR inducing neurons from the VNC could be required to potentiate the response.

These results clearly demonstrate a role for brain neurons in the SP response. However, we noticed that the flipase approach can result in false negatives as *fruflp* inserted in the same position in the endogenous locus as *fruGAL4* does not induce a response with *UAS FRTstopFRT* mSP in contrast to *fruGAL4*-induced expression of mSP. In contrast, the same experiment with *dsxGAL4* and *dsxflp* results in a positive SP response indistinguishable from mated females (*Haussmann et al., 2013*).

Next, we tested whether neuronal activation or inhibition would induce a PMR. Strikingly, conditional activation of *SPR8 ∩ dsx*, *fru11/12 ∩ dsx*, or *SPR8 ∩ fru11/12* brain neurons with TrpA1 in adult females completely inhibited receptivity and induced egg laying comparable to mated females (*Figure 7C and D*). In contrast, inhibition of these neurons with tetanus toxin (TNT) did not alter the virgin state, for example, receptivity was not reduced and egg laying was not induced (*Figure 7C and D*).

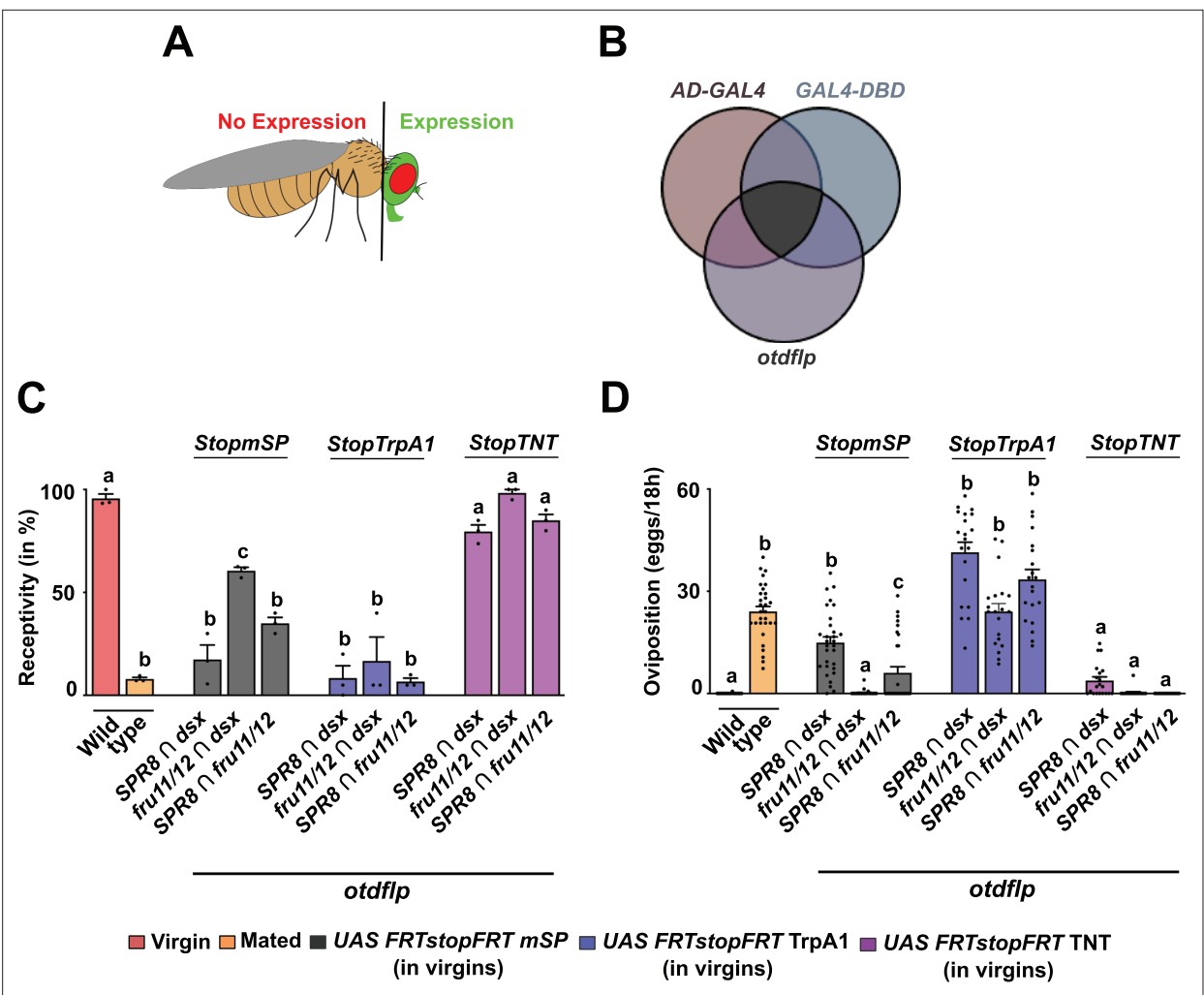

**Figure 7.** Distinct neuronal circuitries in the brain sense SP to induce post-mating responses (PMRs). (**A, B**) Schematic depiction of *UAS GFP* (green) expression in the head of *Drosophila* (**A**) combining *split-GAL4* intersectional expression (*AD-GAL4* and *GAL4-DBD*) with brain-expressed *otdflp* mediated recombination of *UAS FRTGFPstopFRTmSP* (**B**). (**C, D**) Receptivity (**C**) and oviposition (**D**) of wild type control virgin (red) and mated (orange) females, and virgin females expressing *UAS FRTGFPstopFRTmSP* (grey), *UAS FRTGFPstopFRTTrpA1* (purple) and *UAS FRTGFPstopFRTTNT* (pink) under the control of *split-GAL4* intersecting *SPR8 ∩ dsx, fru11/12 ∩ dsx* and *SPR8 ∩ fru11/12* patterns with brain-specific FRT-mediated recombination by *otdflp* shown as means with standard error from three repeats for receptivity (21 females per repeat) by counting the number of females mating within a 1-hour period or for oviposition by counting the eggs laid within 18 hours from 30 females. Statistically significant differences from ANOVA post hoc comparison are indicated by different letters ($p < 0.0001$ except $p < 0.0004$ for c in **C**, $p < 0.007$ for c in **D**).

The online version of this article includes the following source data for figure 7:

**Source data 1.** Quantitative results used to generate graphs in *Figure 7C and D*.

## Overlapping expression of *SPSN* with *SPR8* and *dsx* mediates changes in receptivity and oviposition by mSP expression in the brain

When we analysed *split-GAL4* combinations of SPSN (VT058873, the common line in the SPSN1 and 2 lines) with *SPR8*, *fru11/12*, and *dsx*, we observed full response to *mSP* expression for the intersection with *SPR8* and *fru11/12*, and a partial response for the *SPSN ∩ dsx* intersection (*Figure 8A and B*). Intriguingly, all of these *split-Gal4* combinations expressed in few neurons in the brain, the VNC and genital tract neurons, except for VT058873 ∩ fru11/12, which did not express in genital tract neurons (*Figure 8C–N*).

We then restricted expression of mSP and induction of neuronal activity to the head with these split-GAL4 combinations using *FRTstop* cassettes and *otdflp*. In this set-up, we can induce PMRs from mSP expression or neuronal activation from TrpA1 expression with the *VY058873 ∩ SPR8* and *dsx*

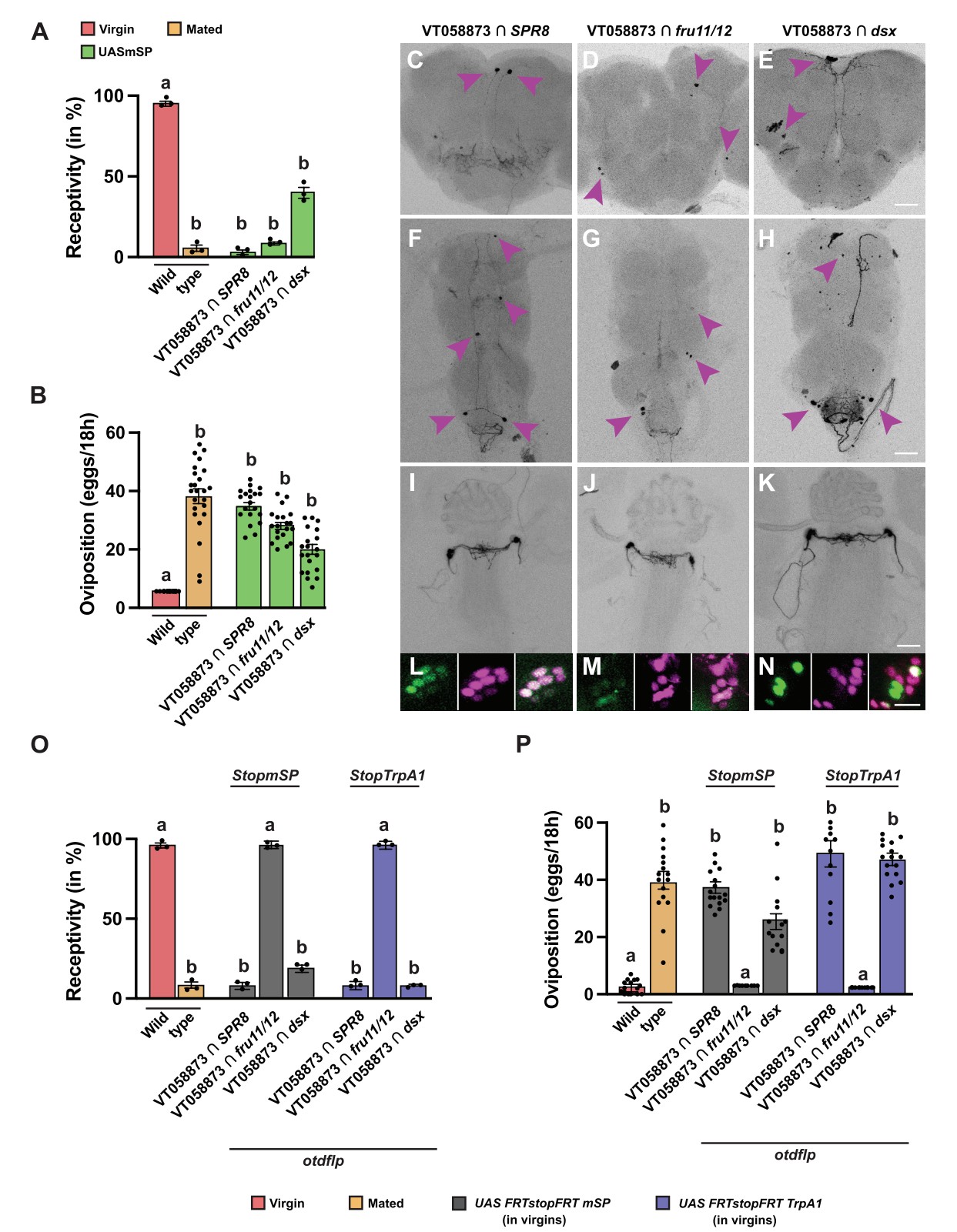

**Figure 8.** Expression of membrane-tethered sex peptide (mSP) in *SPSN VT058873 AD* intersected with *SPR8 DBD, fru11/12 DBD,* and *dsx DBD* induces post-mating responses (PMRs) and *SPSN VT058873 AD* intersected with *SPR8 DBD* and *dsx DBD* sense sex peptide (SP) in the brain. (**A, B**) Receptivity (**A**) and oviposition (**B**) of wild type control virgin (red) and mated (orange) females, and virgin females expressing *UAS mSP* (green) under the control of *VT058873 ∩ SPR8, VT058873 ∩ fru11/12,* and *VT058873 ∩ dsx* shown as means with standard error from three repeats for receptivity (21 females per

*Figure 8 continued on next page*

*Figure 8 continued*

repeat) by counting the number of females mating within a 1-hour period or for oviposition by counting the eggs laid within 18 hours from 30 females. Statistically significant differences from ANOVA post hoc comparison are indicated by different letters (p<0.0001). (**C–H**) Adult female brains (**C–E**) and ventral nerve cords (VNC, **F–H**) expressing *UAS CD8GFP*. Scale bars shown in (**E, H, K, N**) are 50 μm, 100 μm, 100 μm and 20 μm, respectively. (**I–N**) Representative genital tracts labelled with UAS CD8 GFP and genital tract neurons labelled with *UAS H2BYFP* and *elavLexA AopNLStomato*. (**O, P**) Receptivity (**O**) and oviposition (**P**) of wild type control virgin (red) and mated (orange) females, and virgin females expressing *UAS FRTGFPstopFRTmSP* (grey) and *UAS FRTGFPstopFRTTrpA1* (purple) under the control of *split-GAL4* intersecting *VT058873 ∩ SPR8, VT058873 ∩ fru11/12,* and *VT058873 ∩ dsx* patterns with brain-specific FRT-mediated recombination by *otdflp* shown as means with standard error from three repeats for receptivity (21 females per repeat) by counting the number of females mating within a 1-hour period or for oviposition by counting the eggs laid within 18 hours from 30 females. Statistically significant differences from ANOVA post hoc comparison are indicated by different letters (p<0.0001).

The online version of this article includes the following source data and figure supplement(s) for figure 8:

**Source data 1.** Quantitative results used to generate graphs in *Figure 8A, B, O, and P*.

**Figure supplement 1.** Expression analysis of *split-GAL4* in the genital tract.

combination, but not with the *fru11/12* combination (*Figure 8O and P*). For the VT058873 ∩ fru11/12 intersection, PMR inducing neurons likely reside in the VNC.

We then analysed co-expression of *SPR*, *dsx,* and *fru* with SPSN originating *split-GAL4* enhancer lines from *CG31637* (*FD6*), *ocelliless* (*VT05573*), and *Gyc76C* (*VT033490*) in the single-cell brain atlas (*Li et al., 2022*). *CG31637* co-expressed in many cells with *SPR* and *fru*, but only a few cells with *dsx* (*Figure 8—figure supplement 1A–C*). Expression of *ocelliless* with *SPR* and *fru* is broad, while only one neuron expressed with and *Gyc76C* in the brain (*Figure 8—figure supplement 1D, E, G, and H*). Expression of *ocelliless* with *dsx* is restricted to two neurons, and no overlap was detected with *Gyc76C* in the brain (*Figure 8—figure supplement 1F and I*).

### *ppk* neurons do not intersect with SPR, fru, dsx, and FD6 neurons in inducing PMRs by mSP

Expression of *UASmSP* using a *GAL4* driven by a promoter fragment of the *ppk* gene can also induce PMRs (*Figure 9A and B*; *Häsemeyer et al., 2009*; *Yang et al., 2009*). The complement of neurons labelled with *ppkGAL4* consists of at least two populations including prominently sensory neurons, but also eight interneurons in the central brain (*Nallasivan et al., 2021*). These brain neurons show severe developmental defects in SP-insensitive *Nup54* mutant alleles, but they receive inhibitory input from sensory neurons (*Nallasivan et al., 2021*).

To evaluate whether *ppkGAL4* neurons are part of the previously identified expression patterns, we intersected them by crossing *GAL4-AD* lines *SPR8, SPR12* and *fru11/12* and the pan-neural *nSybAD* with a *ppk GAL4-DBD* line containing the previously used 3 kb promoter fragment (*Grueber et al., 2003*; *Seidner et al., 2015*; *Riabinina et al., 2019*). Surprisingly, none of these *split-GAL4* combinations reduced female receptivity or increased egg laying (*Figure 9A and B*).

Few GFP-expressing neurons were detected in the brain for the *nSyb ∩ ppk* and the *fru11/12 ∩ ppk* intersection (*Figure 9C–F*) or abdominal ganglion (*Figure 9G–J*). For the *nSyb ∩ ppk* and the *SPR8 ∩ ppk* intersection, we detected GFP expression in genital tract sensory neurons (*Figure 9K, L, O, and P*), but not for the other combinations (*Figure 9M, N, Q, and R*).

Inhibiting or activating neurons with these *split-Gal4* combinations did not reduce receptivity or induce egg laying (*Figure 9S–V*). How exactly *ppk* neurons labelled with *ppkGAL4* impact on PMRs, however, needs to be further evaluated in follow-up studies. Moreover, if genetical tract neurons were SP target sites, an SP response would have been expected for the *nSyb ∩ ppk* intersection, which we did not observe.

### Female post-mating neuronal circuitry contains neurons that reduce receptivity without inducing oviposition in response to mSP

A number of additional *split-GAL4* combinations with restricted expression have been identified that play a role in female reproductive behaviours (*Wang et al., 2020a*; *Wang et al., 2020b*; *Wang et al., 2021*). These lines express in a subset of *dsx* expressing neurons (*pC1-SS1*), in oviposition descending neurons (*oviDN-SS1* and *2*), in oviposition excitatory neurons (*oviEN-SS1* and *2*), in oviposition inhibitory neurons (*oviIN-SS1* and *2*), and in vaginal plate opening neurons (*vpoDN-SS1*, also termed

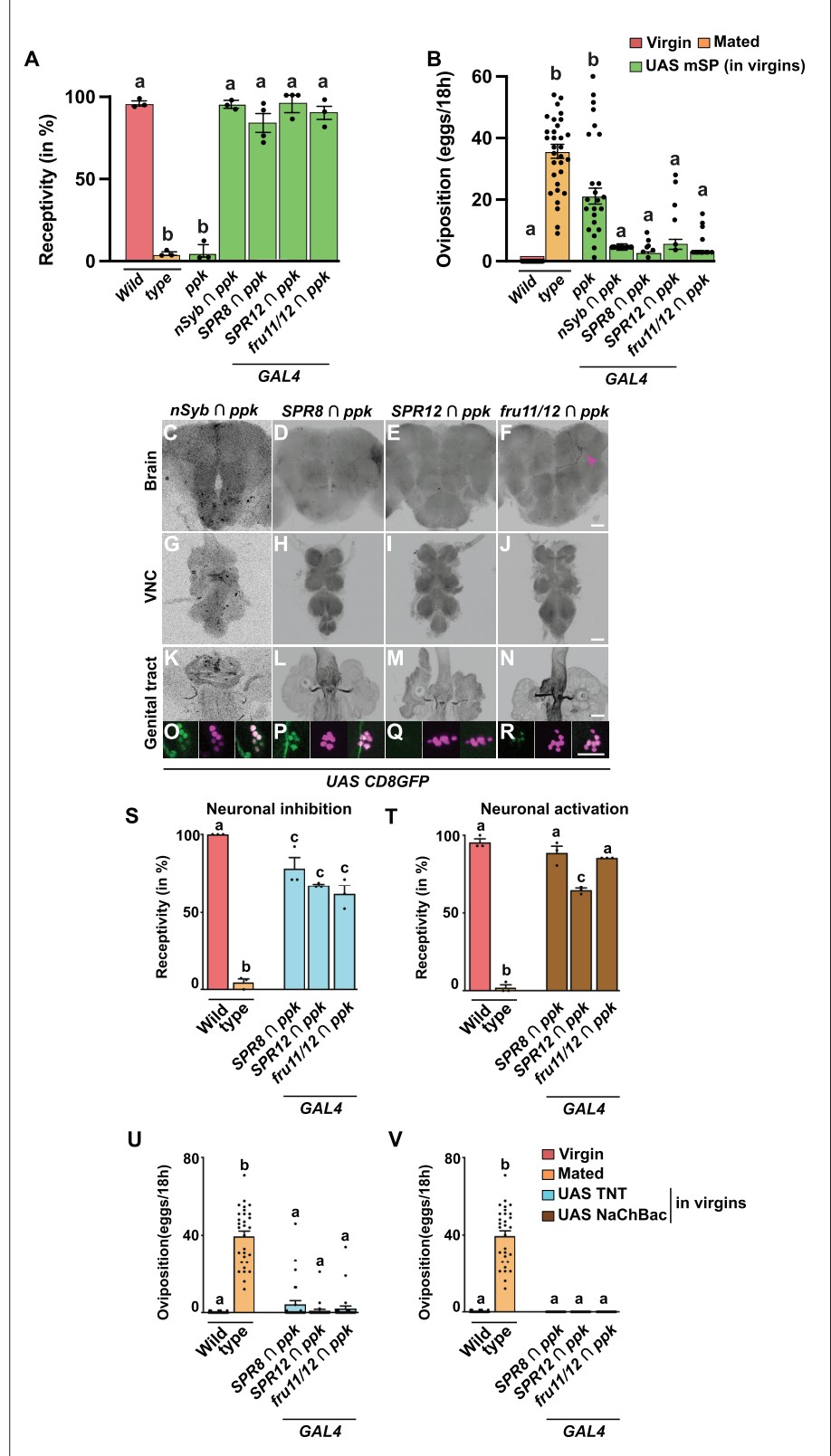

**Figure 9.** *ppk* is not part of the *SPR8, SPR12,* and *fru11/12* post-mating response (PMR)-inducing neuronal circuitry. (**A, B**) Receptivity (**A**) and oviposition (**B**) of wild type control virgin (red) and mated (orange) females, and virgin females expressing *UAS mSP* (green) under the control of *GAL4* in *ppk* or in *nSyb ∩ ppk, SPR8 ∩ ppk, SPR12 ∩ ppk,* and *fru11/12 ∩ ppk* patterns shown as means with standard error from three repeats for receptivity (21 females

*Figure 9 continued on next page*

*Figure 9 continued*

per repeat) by counting the number of females mating within a 1-hour period or for oviposition by counting the eggs laid within 18 hours from 30 females. Statistically significant differences from ANOVA post hoc comparison are indicated by different letters (p<0.0001). (**C–R**) Representative adult female brains, ventral nerve cords (VNC) and genital tracts expressing *UAS CD8GFP* under the control of *UAS* by *nSyb ∩ ppk, SPR8 ∩ ppk, SPR12 ∩ ppk,* and *fru11/12 ∩ ppk*. Scale bars shown in (**F, J, N, R**) are 50 µm, 100 µm, 100 µm and 20 µm, respectively. (**S–V**) Receptivity (**S, T**) and oviposition (**U, V**) of wild type control virgin (red) and mated (orange) females, and virgin females expressing either *UAS TNT* (azure) or *UAS NaChBac* (brown) to inhibit or activate neurons in *SPR8 ∩ ppk, SPR12 ∩ ppk,* and *fru11/12 ∩ ppk* patterns shown as means with standard error from three repeats for receptivity (21 females per repeat) by counting the number of females mating within a 1-hour period or for oviposition by counting the eggs laid within 18 hours from 30 females. Statistically significant differences from ANOVA post hoc comparison are indicated by different letters (p<0.001 for b, and p<0.01 for c in **L** and **N**).

The online version of this article includes the following source data for figure 9:

**Source data 1.** Quantitative results used to generate graphs in *Figure 9A and B*.

ovipositor extrusion/rejection behaviour neurons, because *Drosophila* does not have vaginal plates like e.g. seen in *Hemiptera*; *Aigaki et al., 1991*; *Soller et al., 2006*). When we analysed these lines for a response to mSP expression, receptivity was reduced from mSP expression in *oviEN-SS2, oviN-SS1,* and *vpoDN-SS1* neurons, but no egg laying was induced from mSP expression in any of these neurons (*Figure 10A and B*).

In genital tract neurons, *OviDN-SS1s, OviEN-SS1, OviIN-SS1,* and *vpoDNs* express, but *OviDN-SS1s* and *OviEN-SS1* express weakly (*Figure 10C and J*).

## Interference with neuronal activity in SPRINz reveals regulatory hierarchy

Both inhibitory and activating neurons have been attributed to impact on PMRs (*Kvitsiani and Dickson, 2006*; *Yapici et al., 2008*; *Rezával et al., 2012*). These neurons seem to be part of intersecting circuitry as general inhibition of *ppkGAL4* neurons by TNT only partially blocks the SP response in contrast to inhibition of *ppkGAL4* neurons in the brain alone (*Nallasivan et al., 2021*).

When we inhibited neuronal activity by expression of TNT (*Sweeney et al., 1995*), we observed a significant reduction of receptivity for all *split-Gal4* combinations, though only partially for inhibition in *fru11/12 ∩ FD6* neurons. Likewise, all *split-Gal4* combinations induced a significant increase in egg laying (*Figure 11A and B*). Ablation of these neurons by expression of apoptosis-inducing *reaper* and *hid* genes essentially replicated the results from neuronal inhibition indicating that SPR target neurons are modulatory and are not part of motor circuits because females laid eggs and performed normally in receptivity assays (*Figure 11C and D*).

To evaluate the composition of the intersected expression patterns into inhibitory and activating neurons, we also expressed the *Bacillus halodurans* sodium channel (NaChBac) (*Feng et al., 2014*) to activate all of the intersected neurons. Here, we found a significant reduction of receptivity for four of the five *split-GAL4* combinations, though only partially for activation of *SPR8 ∩ dsx* neurons (*Figure 11E*). Activating *fru11/12 ∩ FD6* neurons did not reduce receptivity (*Figure 11E*). Likewise, we found the same pattern for the induction of egg laying (*Figure 11F*). Four of the five *split-GAL4* combinations induced a significant increase which was only partial in *SPR8 ∩ dsx* neurons, and no egg laying was induced by activating *fru11/12 ∩ FD6* neurons.

Essentially, these results are consistent with previous findings that inhibitory neurons prevail (*Nallasivan et al., 2021*), possibly as input from trunk neurons as found for *ppk* expressing neurons.

## mSP-responsive neurons operate in higher order sensory processing in the brain

With the *split-GAL4* approach, we identified five distinct neuronal sub-types that can induce PMRs. To find out whether these neurons receive input from distinct entry points in the brain and to identify the target neurons of these mSP-responsive neurons, we used the *retro-* and *trans-*Tango technique to specifically activate reporter gene expression in up- and down-stream neurons (*Talay et al., 2017*; *Sorkaç et al., 2023*; *Figure 12A–O*).

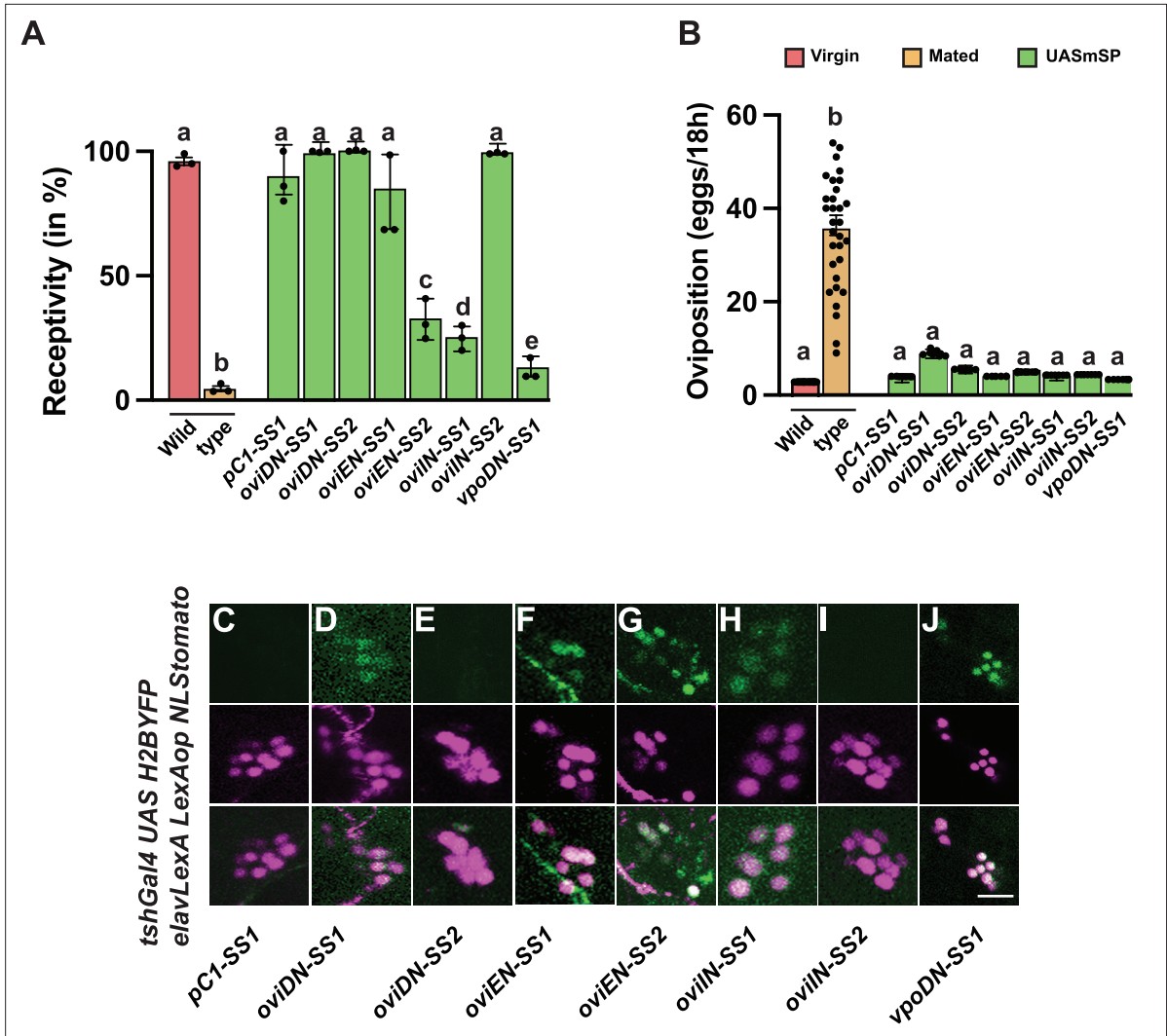

**Figure 10.** Expression of membrane-tethered sex peptide (mSP) in female reproductive behaviour regulating neuron *split-GAL4* lines. (**A, B**) Receptivity (**A**) and oviposition (**B**) of wild type control virgin (red) and mated (orange) females, and virgin females expressing *UAS mSP* (green) under the control of *pC1-SS1, oviDN-SS1 and 2, oviEN-SS1 and 2, oviIN-SS1 and 2, and vpoDN-SS1* shown as means with standard error from three repeats for receptivity (21 females per repeat) by counting the number of females mating within a 1-hour period or for oviposition by counting the eggs laid within 18 hours from 30 females. Statistically significant differences from ANOVA post hoc comparison are indicated by different letters (p<0.0001). (**C–J**) Representative genital tract neurons labelled with *UAS H2BYFP* and *elavLexA AopNLStomato*. The scale bar shown in (**J**) is 20 µm.

The online version of this article includes the following source data for figure 10:

**Source data 1.** Quantitative results used to generate graphs in *Figure 10A and B*.

In the brain, the *retro*-Tango analysis did not identify primary sensory neurons, but higher order neurons in the central brain in all five *split-GAL4* combinations (*Figure 12A–E*). In addition, neurons in the suboesophageal ganglion were marked from *SPR8* intersections with *dsx* and *FD6*, and in *dsx ∩ fru11/12*. In *dsx ∩ fru11/12*, neurons in the optic lobe (medulla) were marked. In addition, a strong signal was observed in all five *split-GAL4* combinations in the mushroom bodies (*Figure 12A–E*). Although mushroom bodies are dispensable for PMRs (*Fleischmann et al., 2001*), their connection to SP target neurons indicates an experience-dependent component of PMRs.

The *trans*-Tango analysis identified a subset of neurons with cell bodies in the suboesophageal ganglion with projections to the *pars intercerebralis* for *SPR8 ∩ dsx* and *fru11/12 ∩ dsx* neurons (*Figure 12K and L*). For *SPR8 ∩ fru11/12* and *SPR8 ∩ FD6* neurons, common target neurons were found in the antennal mechanosensory and motor centre (AMMC) region with a single neuron

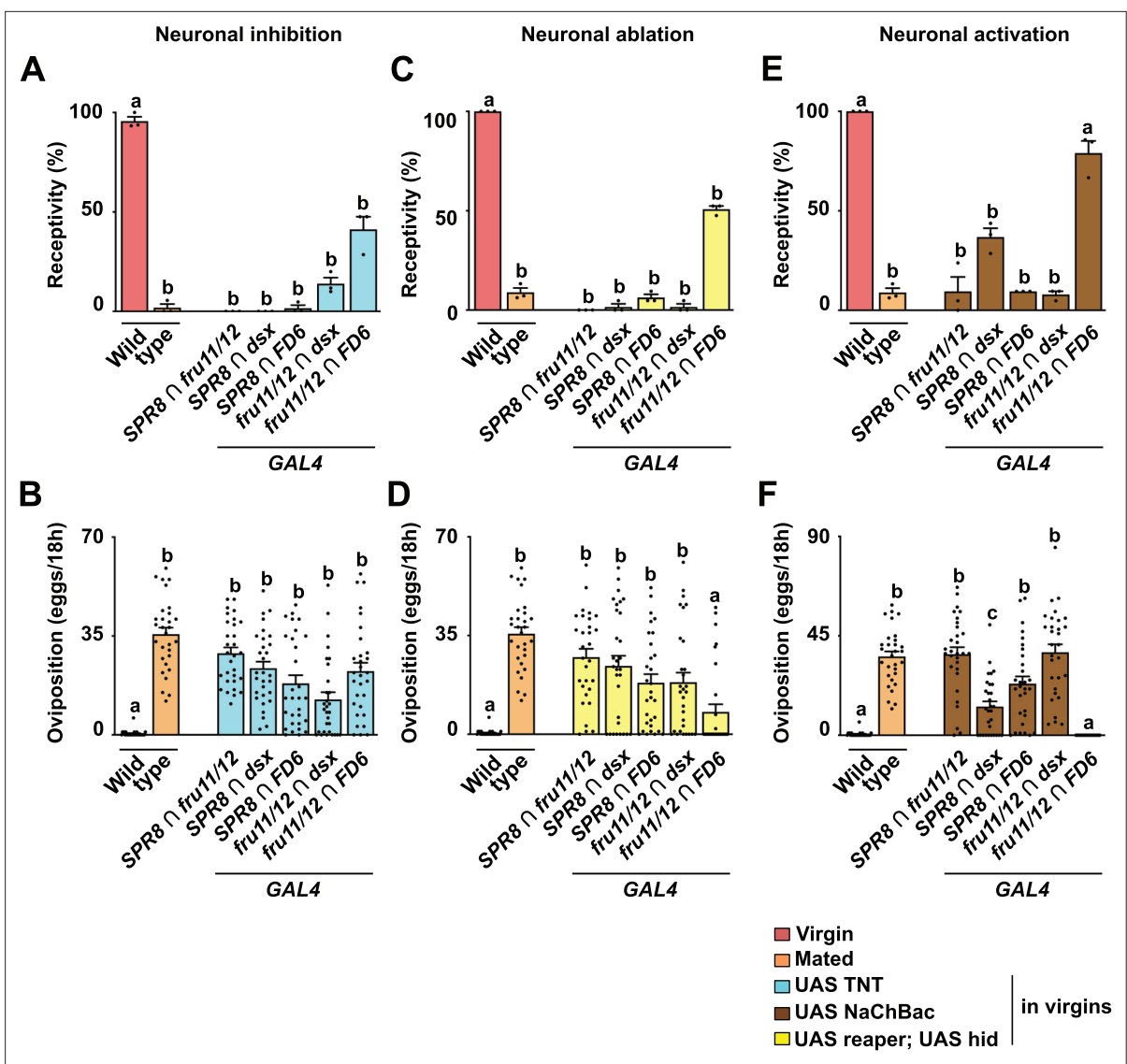

**Figure 11.** Post-mating responses (PMRs) after neuronal inhibition, ablation, or activation of distinct circuits from intersection of *SPR*, *fru*, *dsx*, and *FD6* patterns in the brain and ventral nerve cord (VNC). (**A–F**) Receptivity (**A, C, E**) and oviposition (**B, D, F**) of wild type control virgin (red) and mated (orange) females, and virgin females expressing either *UAS TNT* (azure, **A, B**) or *UAS reaper hid* to inhibit or ablate neurons (yellow, **C, D**), respectively, or *UAS NaChBac* (brown, **E, F**) to activate neurons in *SPR8 ∩ fru11/12, SPR8 ∩ dsx, SPR8 ∩ FD6, fru11/12 ∩ dsx*, and *fru11/12 ∩ FD6 split-Gal4* patterns shown as means with standard error from three repeats for receptivity (21 females per repeat) by counting the number of females mating within a 1-hour period or for oviposition by counting the eggs laid within 18 hours from 30 females. Statistically significant differences from ANOVA post hoc comparison are indicated by letters (p≤0.0095 in **A, B**, p<0.0001 in **C, D** except p=0.016 for c in **D**, p<0.0001 in **E** and p<0.0002 in **F**).

The online version of this article includes the following source data for figure 11:

**Source data 1.** Quantitative results used to generate graphs in *Figure 11A–F*.

identified near the mushroom body region (*Figure 12M and N*; *Ishimoto and Kamikouchi, 2021*). For *fru11/12 ∩ FD6*, no obvious targets were identified in the central brain (*Figure 12O*).

In the VNC, the *trans*-Tango analysis showed post-synaptic targets within the abdominal ganglion with all five *split-GAL4* combinations indicating an interconnected neuronal network (*Figure 12— figure supplement 1A–O*), which needs to be elaborated in detail. In the genital tract, no post-synaptic targets were detected, indicating that these are afferent neurons integrating sensory input (*Figure 12—figure supplement 1P–AD*).

Taken together, circuitries identified via *retro*- and *trans*-Tango place SP target neurons at the interface of sensory processing interneurons connecting to two commonly shared post-synaptic

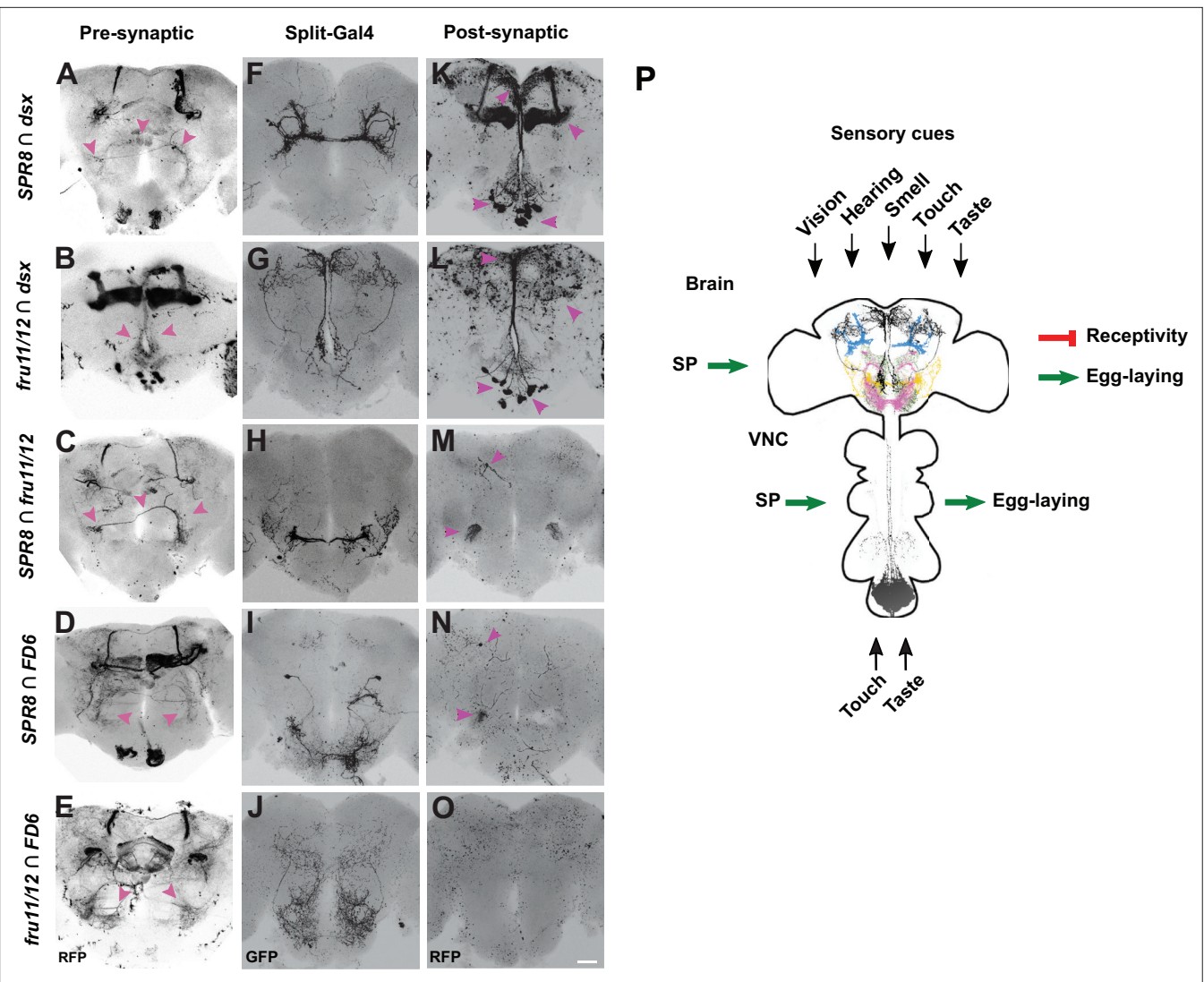

**Figure 12.** *retro-* and *trans-*Tango identification of pre- and post-synaptic neurons of SP target neurons reveals higher order neuronal input canalised into shared output circuitries. (**A–O**) Representative adult female brains expressing *QUAST tomato3xHA retro-*Tango (left, **A–E**), *UAS myrGFP* (middle, **F–J**) and *QUAST tomato3xHA trans-*Tango (right, **K–O**) in *SPR8 ∩ dsx, fru11/12 ∩ dsx, SPR8 ∩ fru11/12, SPR8 ∩ FD6*, and *fru11/12 ∩ FD6 split-GAL4s*. The presynaptic (**A–E**, left), *split-GAL4* (**F–J**, middle) and postsynaptic (**K–O**, right) neuronal circuitries are shown in an inverted grey background. Arrows (magenta) indicate neurons and their corresponding projections in different regions in the female brain. The scale bar shown in (**O**) is 50 µm. (**P**) Model for the SP induced post-mating response. SP interferes with interpretation of sensory cues, for example, vision, hearing, smell, taste, and touch at distinct sites in the brain indicated by higher order projections revealed by intersectional expression in the following patterns: *SPR8 ∩ dsx* (blue), *fru11/12 ∩ dsx* (black), *SPR8 ∩ fru11/12* (yellow), *SPR8 ∩ FD6* (pink), and *fru11/12 ∩ FD6* (olive). and VNC (*fru11/12 ∩ dsx*) during higher order neuronal processing.

The online version of this article includes the following figure supplement(s) for figure 12:

**Figure supplement 1.** *trans-*Tango identifies post-synaptic proceeding neurons of sex peptide (SP) targets in the ventral nerve cord (VNC), but not the genital tract.

processing neuronal populations in the brain. Hence, our data indicate that SP interferes with sensory input processing from multiple modalities that are canalised to higher order processing centres to generate a behavioural output.

## Discussion

Much has been learned about the neuronal circuitry governing reproductive behaviours in *Drosophila* from interfering with neuronal activity in few neurons selected by intersectional expression using

*split-GAL4* (*Wang et al., 2020a*; *Wang et al., 2020b*; *Wang et al., 2021*). However, how SP signalling as main inducer of the PMR, prominently consisting of refractoriness to re-mate and induction of egg laying, is integrated in this circuitry is not completely understood (*Haussmann et al., 2013*).

Here, we addressed this gap by identifying regulatory regions in *SPR*, *fru,* and *dsx* genes driving membrane-tethered expression of SP in subsets of neurons to delineate SP targets to very few neurons in the central brain and the VNC by intersectional expression. Consistent with previous analysis describing multiple pathways for the SP response (*Haussmann et al., 2013*), we find five distinct populations of interneurons in the central brain directing PMRs. In SP target neurons in the central brain, SPR is essential to induce PMRs when receiving SP from males through mating. From mapping post-synaptic targets by *trans*-Tango, we identified two populations of interneurons. The architecture of this circuitry is reminiscent of processing of sensory input transmitted to central brain pattern generators for behavioural output. Hence, SP interferes at several levels for coordinating PMRs, but also leaves the female the opportunity to interfere under unfavourable conditions with specific elements of PMRs, for example, if there is no egg laying substrate, females will still not remate (*Haussmann et al., 2013*). Likewise, mated females will not lay eggs despite suitable egg laying substrates if parasitoid wasps are present (*Kacsoh et al., 2015*). Thus, the architecture of female PMRs contrasts with male-courtship behaviour consisting of a sequel of behavioural elements that once initiated will always follow stereotypically to the end culminating in mating, or start from the beginning when interrupted (*Hall, 1994*; *Greenspan and Ferveur, 2000*).

## SP induces PMRs via entering the hemolymph to target neurons in the central brain and ventral nerve cord

Early characterisation of the SP signalling cascade demonstrated induction of PMRs from various other sources than mating, including transgenic secretion from the fat body, expression as membrane-tethered form on neurons or injection of synthetic peptide into the hemolymph (*Chen et al., 1988*; *Aigaki et al., 1991*; *Schmidt et al., 1993*; *Nakayama et al., 1997*). Likewise, SP is detected in the haemolymph after mating at a PMR inducing concentration (*Haussmann et al., 2013*). Moreover, PMRs are induced faster when SP is injected compared to induction by mating (*Haussmann et al., 2013*). This delay, however, is not attributed to sperm binding of SP as it is unchanged after mating with spermless males. These results suggest that SP reaches its targets through entering the circulatory system to target neurons and contrasts a previously proposed model favouring genital tract neurons as SP sensors from the lumen of the genital tract (*Häsemeyer et al., 2009*; *Yang et al., 2009*; *Rezával et al., 2012*). We previously observed binding of radiolabelled SP to various sites in the nervous system including afferent nerves, but these signals likely reflect binding to broadly expressed SPR rather than binding to SPRINz (*Ottiger et al., 2000*; *Ding et al., 2003*; *Yapici et al., 2008*; *Haussmann et al., 2013*).

In further support of the internalisation model, we identified *GAL4* drivers that express mSP in genital tract neurons, but do not induce PMRs. Also, *SPR12* does not express in genital tract neurons, but induces egg laying by expression of mSP. Moreover, expression of mSP predominantly in the trunk (including all genital tract sensory neurons) only induces egg laying, but does not change receptivity. Likewise, expression of mSP specifically in the brain (*SPR8 ∩ dsx*) can reduce receptivity and induce egg laying indistinguishable from mated females.

A *ppkGAL4* line generated by P-element mediated transformation can induce PMRs by expression of *UAS mSP* (*Grueber et al., 2003*). The same promoter fragment fused to a *GAL4 DBD* and inserted by phiC31 integration into a landing site intersected with pan-neural *nSyb AD* line (*Seidner et al., 2015*; *Riabinina et al., 2019*), however, does not induce an SP response despite being expressed in genital tract neurons. We found that the *ppkGAL4* expresses in a few neurons in the brain and VNC (*Nallasivan et al., 2021*), but this expression is absent in *nSyb ∩ ppk* intersection. Likely, the *ppkGAL4* construct is inserted in a locus that contains an enhancer that drives expression in SP target neurons.

These results are in strong favour of SP entering the hemolymph to target neurons in the VNC for inducing egg laying, and in the central brain for reducing receptivity and inducing egg laying (*Haussmann et al., 2013*).

## Integration of SP signalling into the circuitry directing reproductive behaviours

Reduction of receptivity and induction of egg laying are both induced by the same critical concentration of injected SP (*Schmidt et al., 1993*; *Haussmann et al., 2013*) initially suggesting a simple on/off system for PMRs likely initiated from a small population of neurons. However, such a model would not allow us to split the SP response into individual PMR components by expression of mSP.

Here, we identified several *GAL4* drivers, which can induce only egg laying (*SPR12*, *FD3*, *FD4*, and *tsh GAL4*), but do not reduce receptivity, and others that can only reduce receptivity (*oviEN-SS2*, *oviIN-SS1*, and *vpoDN-SS1*), but do not induce egg laying. Strikingly, *tshGAL4*, which expresses predominantly in the trunk, only affects egg laying, suggesting a role for the abdominal ganglion in egg laying. Moreover, *dsx* and all of the SBRINz *split-GAL4* combinations affect egg laying and express in the abdominal ganglion (*Rezával et al., 2012*; *Zhou et al., 2014*). Hence, this neuronal structure has a key role in regulating egg laying. Since more than a single neuronal population seems to direct egg laying, further high-resolution mapping is required to identify individual neuronal population within the abdominal ganglion (*Jang et al., 2017*; *Oliveira-Ferreira et al., 2023*).

Since *tshGAL4* only induces egg laying, neurons in the brain must direct reduction of receptivity. Through intersectional expression in combination with head-specific expression of *otdflp*, we could express mSP only in the brain by FLP-mediated brain-specific excision of a stop cassette. We observed a significant reduction in receptivity for all five intersections tested, but for four, the response is only partial, likely due to the inefficiency of FLP-mediated recombination.

Moreover, brain neurons can also induce egg laying when *SPR8* is intersected with *dsx*, and to some extent also from *SPR8* intersection with *fru11/12*. Due to the inefficiency of FLP-mediated recombination, however, this is likely an underestimate and solving this issue requires development of more robust tools.

In any case, however, our results show that PMRs can be induced from mSP expression from several sites, suggesting interference with processing of sensory information at the level of interneurons. In particular, *SPR8* ∩ *fru11/12* neurons resemble auditory AMMC-B2 neurons involved in processing of information of the male love song (*Yamada et al., 2018*). Likewise, *SPR8* ∩ *dsx* neurons seem to overlap with dimorphic *dsx* pCL2 interneurons that are part of the 26 neurons constituting the pC2 neuronal population involved in courtship song sensing, mating acceptance and ovipositor extrusion for rejection of courting males (*Kimura et al., 2015*; *Deutsch et al., 2019*; *Wang et al., 2020a*). The *SPR8* ∩ *FD6* neurons resemble dopaminergic *fru* P1 neurons involved in courtship and the *fru11/12* ∩ *dsx* neurons seem to overlap with *dsx* pCd and neuropeptide F neurons involved in courtship (*Zhang et al., 2021*). In females, pC1d neurons have been linked to aggression (*Deutsch et al., 2020*; *Schretter et al., 2020*). The *fru11/12* ∩ *FD6* neurons resemble a class of gustatory pheromone sensing neurons (*Sakurai et al., 2013*). Although we likely have not identified all SP sensing neurons, our resources will provide a handle to future exploration of the details of this neuronal circuitry incorporating SP signalling for inducing PMRs.

### Conclusions

We have identified distinct SP response-inducing neurons (SPRINz) in the central brain and the VNC. Since these five different SP response-inducing neuronal populations in the central brain converge into two target sites, our data suggest a model (*Figure 12P*), whereby SP signalling interferes with integration of sensory input. Independent interference with different sensory modalities opts for the female to counteract male manipulation at the level of perception of individual sensory cues to adapt to varying physiological and environmental conditions to maximise reproductive success.

## Materials and methods

**Key resources table**

| Reagent type (species) or resource | Designation | Source or reference | Identifiers | Additional information |
|---|---|---|---|---|
| Genetic reagent (*Drosophila melanogaster*) | Wild-type: Canton S | This study | RRID:BDSC_64349 | Wild-type strain |

*Continued on next page*

*Continued*

| Reagent type (species) or resource | Designation | Source or reference | Identifiers | Additional information |
|---|---|---|---|---|
| Genetic reagent (*D. melanogaster*) | w*; UASmSP (3<sup>rd</sup>, 61C) | *Nakayama et al., 1997* | | Gift from T. Aigaki |
| Genetic reagent (*D. melanogaster*) | dsx-GAL4 inserted into the endogenous dsx gene (84E5-84E6) | *Rideout et al., 2010* | | Gift from S. Goodwin |
| Genetic reagent (*D. melanogaster*) | fru-GAL4 inserted into the endogenous fru gene (91A6-91B3) | Dickson lab | RRID:BDSC_66870 | |
| Genetic reagent (*D. melanogaster*) | nSyb GAL4 (3<sup>rd</sup>) | *Rezával et al., 2012* | | Gift from S. Goodwin |
| Genetic reagent (*D. melanogaster*) | ppk-GAL4/CyO | Bloomington Stock Centre | RRID:BDSC_49021 | |
| Genetic reagent (*D. melanogaster*) | tshGAL4-1/CyO | Bloomington Stock Centre | RRID:BDSC_3040 | |
| Genetic reagent (*D. melanogaster*) | elav FRTstopFRT GAL4 | *Zaharieva et al., 2015* | | |
| Genetic reagent (*D. melanogaster*) | otdflp | *Asahina et al., 2014* | | Gift from D. Anderson |
| Genetic reagent (*D. melanogaster*) | UASmCD8GFP (X) | Bloomington Stock Centre | RRID:BDSC_5136 | |
| Genetic reagent (*D. melanogaster*) | UASmCD8GFP (2nd) | Bloomington Stock Centre | RRID:BDSC_5137 | |
| Genetic reagent (*D. melanogaster*) | UAS-H2B::YFP (2<sup>nd</sup>) | *Li et al., 2020* | | Gift from A. Hidalgo |
| Genetic reagent (*D. melanogaster*) | elavLexA (2nd) | Bloomington Stock Centre | RRID:BDSC_52676 | |
| Genetic reagent (*D. melanogaster*) | LexAop NLStomato (2nd) | Bloomington Stock Centre | RRID:BDSC_66690 | |
| Genetic reagent (*D. melanogaster*) | UAS TNT (2nd) | *Sweeney et al., 1995* | | Gift from J.J. Hodge |
| Genetic reagent (*D. melanogaster*) | UAS TrpA1 (3rd) | Bloomington Stock Centre | RRID:BDSC_26264 | |
| Genetic reagent (*D. melanogaster*) | UASFlybow 1.1 (myrGFP, 2nd) | Bloomington Stock Centre | RRID:BDSC_35537 | |
| Genetic reagent (*D. melanogaster*) | UAS-NaCh::BacGFP (3rd) | Bloomington Stock Centre | RRID:BDSC_9467 | |
| Genetic reagent (*D. melanogaster*) | UAS Reaper/FM7;UAS Hid/CyO | Bloomington Stock Centre | RRID:BDSC_5823 | |
| Genetic reagent (*D. melanogaster*) | UAS FRTstopFRT GFP/CyO | Dickson lab | RRID:BDSC_30125 | |
| Genetic reagent (*D. melanogaster*) | UAS FRTstopFRT TNT/CyO | Dickson lab | RRID:BDSC_30125 | |
| Genetic reagent (*D. melanogaster*) | UAS FRTstopFRT TrpA1/CyO | Dickson lab | RRID:BDSC_30125 | |
| Genetic reagent (*D. melanogaster*) | UAS FRTstopFRT mSP (3<sup>rd</sup>) | This study | Soller Lab | *UAS mSP* line with a stop cassette |
| Genetic reagent (*D. melanogaster*) | UAS dicer2; UAS SPR RNAi (X, 3rd) | *Yapici et al., 2008* | | Gift from B. Dickson lab |
| Genetic reagent (*D. melanogaster*) | SPR | Bloomington Stock Centre | RRID:BDSC_7708 | |
| Genetic reagent (*D. melanogaster*) | Df(1)JC70/FM7c | Bloomington Stock Centre | RRID:BDSC_944 | |

*Continued on next page*

*Continued*

| Reagent type (species) or resource | Designation | Source or reference | Identifiers | Additional information |
|---|---|---|---|---|
| Genetic reagent (*D. melanogaster*) | nSyb p65-GAL4.AD (attP40) | *Riabinina et al., 2019* | | Gift from O. Riabinina |
| Genetic reagent (*D. melanogaster*) | SPR8 AD: VT057286-p65. AD (attP40) | Bloomington Stock Centre | RRID:BDSC_71392 | |
| Genetic reagent (*D. melanogaster*) | Fru11/12 AD: VT043695-p65.AD (attP40) | Bloomington Stock Centre | RRID:BDSC_72065 | |
| Genetic reagent (*D. melanogaster*) | dsx DBD | *Rideout et al., 2010* | | Gift from S. Goodwin |
| Genetic reagent (*D. melanogaster*) | dsx24 DBD: R42G02-GAL4. DBD (attP2) | Bloomington Stock Centre | N/A | |
| Genetic reagent (*D. melanogaster*) | SPR8 DBD: VT057286-Gal4.DBD (attP2) | Bloomington Stock Centre | RRID:BDSC_71425 | |
| Genetic reagent (*D. melanogaster*) | fru11/12 DBD: VT043695-GAL4.DBD (attP2) | Bloomington Stock Centre | RRID:BDSC_72788 | |
| Genetic reagent (*D. melanogaster*) | FD6 DBD: VT003280-GAL4.DBD (attP2) | Bloomington Stock Centre | RRID:BDSC_75877 | |
| Genetic reagent (*D. melanogaster*) | ppk DBD: ppk-GAL4.DBD (VK00027, 89E11) | *Seidner et al., 2015* | | Gift from W. J. Joiner |
| Genetic reagent (*D. melanogaster*) | SPR12 AD: VT057292-p65. AD (attP40) | Bloomington Stock Centre | RRID:BDSC_72924 | |
| Genetic reagent (*D. melanogaster*) | UAS-myrGFP QUAS-mtdTomato-3xHA; trans-Tango | Bloomington Stock Centre | RRID:BDSC_95317 | |
| Genetic reagent (*D. melanogaster*) | QUAS-mtdTomato-3xHA; retro-Tango | *Sorkaç et al., 2023* | RRID:BDSC_99661 | Gift from G. Barnea |
| Genetic reagent (*D. melanogaster*) | fru1 GAL4: R23C03-GAL4 (attP2) | Bloomington Stock Centre | RRID:BDSC_49021 | |
| Genetic reagent (*D. melanogaster*) | fru2 GAL4, R22H11-GAL4 (attP2) | Bloomington Stock Centre | RRID:BDSC_48043 | |
| Genetic reagent (*D. melanogaster*) | fru3 GAL4, R21H09-GAL4 (attP2) | Bloomington Stock Centre | RRID:BDSC_49867 | |
| Genetic reagent (*D. melanogaster*) | fru4 GAL4, R23C12-GAL4 (attP2) | Bloomington Stock Centre | RRID:BDSC_49026 | |
| Genetic reagent (*D. melanogaster*) | fru5 GAL4, R22F06-GAL4 (attP2) | Korea *Drosophila* Resource Center | KDRC 11848 | |
| Genetic reagent (*D. melanogaster*) | fru6 GAL4, R23D03GAL4 | Bloomington Stock Centre | N/A | |
| Genetic reagent (*D. melanogaster*) | fru7 GAL4, R22B09-GAL4 | Bloomington Stock Centre | N/A | |
| Genetic reagent (*D. melanogaster*) | fru8 GAL4, R23B12-GAL4 (attP2) | Korea *Drosophila* Resource Centre | KDRC 11849 | |
| Genetic reagent (*D. melanogaster*) | fru9 GAL4, R22A02-GAL4 (attP2) | Bloomington Stock Centre | RRID:BDSC_49868 | |
| Genetic reagent (*D. melanogaster*) | fru10 GAL4, R22C05-GAL4 (attP2) | Bloomington Stock Centre | RRID:BDSC_49301 | |
| Genetic reagent (*D. melanogaster*) | fru11 GAL4, R22C11-lexA (attP40) | Bloomington Stock Centre | RRID:BDSC_52604 | |
| Genetic reagent (*D. melanogaster*) | fru12 GAL4, R22A11-GAL4 (attP2) | Bloomington Stock Centre | RRID:BDSC_48966 | |
| Genetic reagent (*D. melanogaster*) | fru13 GAL4, R23A06-GAL4 (attP2) | Bloomington Stock Centre | RRID:BDSC_49009 | |

*Continued on next page*

*Continued*

| Reagent type (species) or resource | Designation | Source or reference | Identifiers | Additional information |
|---|---|---|---|---|
| Genetic reagent (*D. melanogaster*) | fru14 GAL4, R22C03-GAL4 (attP2) | Korea Drosophila Resource Centre | RRID:KDRC_11868 | |
| Genetic reagent (*D. melanogaster*) | fru15 GAL4, R23B04-GAL4 (attP2) | Bloomington Stock Centre | RRID:BDSC_49016 | |
| Genetic reagent (*D. melanogaster*) | fru16 GAL4, R22C07-GAL4 (attP2) | Bloomington Stock Centre | RRID:BDSC_48975 | |
| Genetic reagent (*D. melanogaster*) | fru17 GAL4, R23C08-GAL4 (attP2) | Korea Drosophila Resource Centre | KDRC 11835 | |
| Genetic reagent (*D. melanogaster*) | fru18 GAL4, R23C07GAL4 | Bloomington Stock Centre | N/A | |
| Genetic reagent (*D. melanogaster*) | fru19 GAL4, R22B10-GAL4 (attP2) | Bloomington Stock Centre | RRID:BDSC_48969 | |
| Genetic reagent (*D. melanogaster*) | fru20 GAL4, R22E10-GAL4 (attP2) | Bloomington Stock Centre | RRID:BDSC_49302 | |
| Genetic reagent (*D. melanogaster*) | fru21 GAL4, R22D11-GAL4 (attP2) | Bloomington Stock Centre | RRID:BDSC_48982 | |
| Genetic reagent (*D. melanogaster*) | fru22 GAL4, R22H07-GAL4 (attP2) | Bloomington Stock Centre | RRID:BDSC_490003 | |
| Genetic reagent (*D. melanogaster*) | fru23 GAL4, R21H02-GAL4 (attP2) | Korea Drosophila Resource Centre | KDRC 11847 | |
| Genetic reagent (*D. melanogaster*) | fru24 GAL4, R23B11-GAL4 (attP2) | Bloomington Stock Centre | RRID:BDSC_49019 | |
| Genetic reagent (*D. melanogaster*) | fru25 GAL4: VT043674-GAL4 (attP2) | Bloomington Stock Centre | N/A | |
| Genetic reagent (*D. melanogaster*) | fru26 GAL4, VT043675-GAL4 (attP2) | Bloomington Stock Centre | N/A | |
| Genetic reagent (*D. melanogaster*) | fru27 GAL4, VT043676-GAL4 (attP2) | Bloomington Stock Centre | N/A | |
| Genetic reagent (*D. melanogaster*) | dsx1 GAL4, R39E06-GAL4 (attP2) | Bloomington Stock Centre | RRID:BDSC_50051 | |
| Genetic reagent (*D. melanogaster*) | dsx2 GAL4, R40A05-GAL4 (attP2) | Bloomington Stock Centre | RRID:BDSC_48138 | |
| Genetic reagent (*D. melanogaster*) | dsx3 GAL4, R40F03-GAL4 (attP2) | Bloomington Stock Centre | RRID:BDSC_47355 | |
| Genetic reagent (*D. melanogaster*) | dsx4 GAL4, R40F04-GAL4 | Bloomington Stock Centre | N/A | |
| Genetic reagent (*D. melanogaster*) | dsx5 GAL4, R41A01-GAL4 | Bloomington Stock Centre | N/A | |
| Genetic reagent (*D. melanogaster*) | dsx6 GAL4, R41D01GAL4 | Bloomington Stock Centre | N/A | |
| Genetic reagent (*D. melanogaster*) | dsx7 GAL4, R41F06-GAL4 (attP2) | Bloomington Stock Centre | RRID:BDSC_47584 | |
| genetic reagent (*D. melanogaster*) | dsx8 GAL4, R42C06-GAL4 (attP2) | Bloomington Stock Centre | RRID:BDSC_50150 | |
| Genetic reagent (*D. melanogaster*) | dsx9 GAL4, R42D02-GAL4 (attP2) | Bloomington Stock Centre | RRID:BDSC_41250 | |
| Genetic reagent (*D. melanogaster*) | dsx10 GAL4, R42D04-GAL4 (attP2) | Bloomington Stock Centre | RRID:BDSC_47588 | |
| Genetic reagent (*D. melanogaster*) | dsx11 GAL4, VT038171-GAL4 (attP2) | Bloomington Stock Centre | N/A | |

*Continued on next page*

*Continued*

| Reagent type (species) or resource | Designation | Source or reference | Identifiers | Additional information |
|---|---|---|---|---|
| Genetic reagent (*D. melanogaster*) | dsx12 GAL4, VT038169-GAL4 (attP2) | Bloomington Stock Centre | N/A | |
| Genetic reagent (*D. melanogaster*) | dsx13 GAL4, VT038167-GAL4 (attP2) | Bloomington Stock Centre | N/A | |
| Genetic reagent (*D. melanogaster*) | dsx14 GAL4, VT038166-GAL4 (attP2) | Bloomington Stock Centre | N/A | |
| Genetic reagent (*D. melanogaster*) | dsx15 GAL4, VT038161-GAL4 (attP2) | Bloomington Stock Centre | N/A | |
| Genetic reagent (*D. melanogaster*) | dsx16 GAL4, VT038159-GAL4 (attP2) | Bloomington Stock Centre | N/A | |
| Genetic reagent (*D. melanogaster*) | dsx17 GAL4, VT038157-GAL4 (attP2) | Bloomington Stock Centre | N/A | |
| Genetic reagent (*D. melanogaster*) | dsx18 GAL4, VT038155-GAL4 (attP2) | Bloomington Stock Centre | N/A | |
| Genetic reagent (*D. melanogaster*) | dsx19 GAL4, VT038151-GAL4 | Bloomington Stock Centre | N/A | |
| Genetic reagent (*D. melanogaster*) | dsx20 GAL4, VT038149-GAL4 (attP2) | Bloomington Stock Centre | N/A | |
| genetic reagent (*D. melanogaster*) | dsx21 GAL4, P{VT038148-GAL4 (attP2) | Bloomington Stock Centre | N/A | |
| Genetic reagent (*D. melanogaster*) | dsx22 GAL4, P{VT038147-GAL4 (attP2) | Bloomington Stock Centre | N/A | |
| genetic reagent (*D. melanogaster*) | dsx23 GAL4, R22H07-GAL4 (attP2) | Bloomington Stock Centre | N/A | |
| Genetic reagent (*D. melanogaster*) | dsx24 GAL4, R21H02-GAL4 (attP2) | Bloomington Stock Centre | N/A | |
| Genetic reagent (*D. melanogaster*) | dsx25 GAL4, R21B01-GAL4 (attP2) | Bloomington Stock Centre | N/A | |
| Genetic reagent (*D. melanogaster*) | SPR1 GAL4, R78F09-GAL4 | Bloomington Stock Centre | N/A | |
| Genetic reagent (*D. melanogaster*) | SPR2 GAL4, R78F11-GAL4 | Bloomington Stock Centre | N/A | |
| Genetic reagent (*D. melanogaster*) | SPR3 GAL4, R78E11-GAL4 | Bloomington Stock Centre | N/A | |
| Genetic reagent (*D. melanogaster*) | SPR4 GAL4, R78E12-GAL4 (attP2) | Bloomington Stock Centre | RRID:BDSC_40002 | |
| Genetic reagent (*D. melanogaster*) | SPR5 GAL4, R78G09-GAL4 (attP2) | Bloomington Stock Centre | RRID:BDSC_40015 | |
| Genetic reagent (*D. melanogaster*) | SPR6 GAL4, R78G08-GAL4 | Bloomington Stock Centre | N/A | |
| Genetic reagent (*D. melanogaster*) | SPR7 GAL4, R78F07-GAL4 (attP2) | Bloomington Stock Centre | RRID:BDSC_47409 | |
| genetic reagent (*D. melanogaster*) | SPR8 GAL4, R78F10-GAL4 (attP2) | Bloomington Stock Centre | RRID:BDSC_40007 | |
| Genetic reagent (*D. melanogaster*) | SPR9 GAL4, R78G02-GAL4 (attP2) | Bloomington Stock Centre | RRID:BDSC_40010 | |
| Genetic reagent (*D. melanogaster*) | SPR10 GAL4, R78G07-GAL4 | Bloomington Stock Centre | N/A | |
| Genetic reagent (*D. melanogaster*) | SPR11 GAL4, R78G04-GAL4 (attP2) | Bloomington Stock Centre | RRID:BDSC_40012 | |

*Continued on next page*

*Continued*

| Reagent type (species) or resource | Designation | Source or reference | Identifiers | Additional information |
|---|---|---|---|---|
| Genetic reagent (*D. melanogaster*) | SPR12 GAL4, R78F05-GAL4 | Bloomington Stock Centre | N/A | |
| Genetic reagent (*D. melanogaster*) | SPR13 GAL4, R78G05-GAL4 (attP2) | Bloomington Stock Centre | RRID:BDSC_41308 | |
| Genetic reagent (*D. melanogaster*) | SPR14 GAL4, R78G06-GAL4 | Bloomington Stock Centre | N/A | |
| Genetic reagent (*D. melanogaster*) | SPR15 GAL4, R78G03-GAL4 (attP2) | Bloomington Stock Centre | RRID:BDSC_40011 | |
| Genetic reagent (*D. melanogaster*) | SPR16 GAL4, R78F06-GAL4 | Bloomington Stock Centre | N/A | |
| Genetic reagent (*D. melanogaster*) | SPR17 GAL4, R78F12-GAL4 | Bloomington Stock Centre | N/A | |
| Genetic reagent (*D. melanogaster*) | SPR18 GAL4, R78F03-GAL4 | Bloomington Stock Centre | N/A | |
| Genetic reagent (*D. melanogaster*) | SPR19 GAL4, R78F01-GAL4 (attP2) | Bloomington Stock Centre | RRID:BDSC_40003 | |
| Genetic reagent (*D. melanogaster*) | SPR20 GAL4, R78G01-GAL4 (attP2) | Bloomington Stock Centre | RRID:BDSC_40009 | |
| Genetic reagent (*D. melanogaster*) | SPR21 GAL4, R78F02-GAL4 (attP2) | Bloomington Stock Centre | N/A | |
| Genetic reagent (*D. melanogaster*) | SPR22 GAL4, R78F08-GAL4 (attP2) | Bloomington Stock Centre | N/A | |
| Genetic reagent (*D. melanogaster*) | FD1 GAL4, VT050405-GAL4 (attP2) | Vienna Drosophila Stock Centre | VDSC | |
| Genetic reagent (*D. melanogaster*) | FD2 GAL4, VT007068-GAL4 (attP2) | Vienna Drosophila Stock Centre | VDSC | |
| Genetic reagent (*D. melanogaster*) | FD3 GAL4, VT045154-GAL4 (attP2), | Vienna Drosophila Stock Centre | VDSC | |
| Genetic reagent (*D. melanogaster*) | FD4 GAL4, VT000454-GAL4 (attP2) | Vienna Drosophila Stock Centre | VDSC | |
| Genetic reagent (*D. melanogaster*) | FD5 GAL4, VT050247-GAL4 (attP2) | Vienna Drosophila Stock Centre | VDSC | |
| Genetic reagent (*D. melanogaster*) | FD6 GAL4, VT003280-GAL4 (attP2) | Vienna Drosophila Stock Centre | VDSC | |
| Genetic reagent (*D. melanogaster*) | SPR8 GAL4, R78F10-GAL4 (attP2); | Bloomington Stock Centre | RRID:BDSC_40007 | |
| Genetic reagent (*D. melanogaster*) | SPR8 AD [VT057286-p65.AD (attP40)]; fru11/12 DBD [VT043695-GAL4.DBD (attP2)] | This study | Soller Lab | Split gal4 combination of *SPR8-AD* and *fru11/12-DBD* |
| Genetic reagent (*D. melanogaster*) | Fru11/12 AD [VT043695-p65.AD (attP40)]; FD6 DBD [VT003280-GAL4.DBD (attP2)] | This study | Soller Lab | Split gal4 combination of *fru11/12-AD* and *FD6-DBD* |
| Genetic reagent (*D. melanogaster*) | SPR8 AD [VT057286-p65.AD (attP40)]; FD6 DBD [VT003280-GAL4.DBD (attP2)] | This study | Soller Lab | Split gal4 combination of *SPR8-AD* and *FD6-DBD* |
| Genetic reagent (*D. melanogaster*) | SPR8 AD [VT057286-p65.AD (attP40)]; dsx DBD (attP2) | This study | Soller Lab | Split gal4 combination of *SPR8-AD* and *dsx-DBD* |

*Continued on next page*

*Continued*

| Reagent type (species) or resource | Designation | Source or reference | Identifiers | Additional information |
|---|---|---|---|---|
| Genetic reagent (*D. melanogaster*) | Fru11/12 AD [VT043695-p65.AD (attP40)]; dsx DBD (attP2) | This study | Soller Lab | Split gal4 combination of *fru11/12-AD* and *dsx-DBD* |
| Genetic reagent (*D. melanogaster*) | VT058873-GAL4.AD (attP40); SPR8 DBD [VT057286-GAL4.DBD(attP2)] | This study | Soller Lab | Split gal4 combination of *SPSN-AD* and *SPR8-DBD* |
| Genetic reagent (*D. melanogaster*) | VT058873-GAL4.AD (attP40); Fru11/12 DBD [VT043696-GAL4.DBD(attP2)] | This study | Soller Lab | Split gal4 combination of *SPSN-AD* and *fru11/12-DBD* |
| Genetic reagent (*D. melanogaster*) | VT058873-GAL4.AD (attP40); dsx DBD (attp2) | This study | Soller Lab | Split gal4 combination of *SPSN-AD* and *dsx-DBD* |
| Genetic reagent (*D. melanogaster*) | nSyb p65-GAL4.AD (attp40); ppk DBD: ppk-GAL4.DBD [VK00027, 89E11] | This study | Soller Lab | Split gal4 combination of *nSYB-AD* and *ppk-DBD* |
| Genetic reagent (*D. melanogaster*) | SAG1, VT050405-GAL4.AD (attP40); VT007068-GAL4.DBD (attP2) | Bloomington Stock Centre | RRID:BDSC_66875 | Split gal4 combination of *SAG1-AD* and *SPSN-DBD* |
| Genetic reagent (*D. melanogaster*) | pC1-SS1, VT2002064-GAL4.AD (attP40); VT008469-GAL4.DBD (attP2) | Bloomington Stock Centre | RRID:BDSC_86830 | |
| Genetic reagent (*D. melanogaster*) | oviDN-SS1, VT050660-GAL4.AD (attP40); VT028160-GAL4.DBD (attP2) | Bloomington Stock Centre | RRID:BDSC_86832 | |
| Genetic reagent (*D. melanogaster*) | oviDN-SS2, VT026873-GAL4.AD (attP40); VT040574-GAL4.DBD (attP2) | Bloomington Stock Centre | RRID:BDSC_86831 | |
| Genetic reagent (*D. melanogaster*) | oviEN-SS1, VT043086-GAL4.AD (attP40); VT034612-GAL4.DBD (attP2) | Bloomington Stock Centre | RRID:BDSC_86839 | |
| Genetic reagent (*D. melanogaster*) | oviEN-SS2, VT034612-GAL4.AD (attP40); VT050229-GAL4.DBD (attP2) | Bloomington Stock Centre | RRID:BDSC_86833 | |
| Genetic reagent (*D. melanogaster*) | oviIN-SS1, R68A10-GAL4.AD (attP40); VT010054-GAL4.DBD (attP2) | Bloomington Stock Centre | RRID:BDSC_86837 | |
| Genetic reagent (*D. melanogaster*) | oviIN-SS2, VT026347-GAL4.AD (attP40); VT026035-GAL4.DBD (attP2) | Bloomington Stock Centre | RRID:BDSC_86838 | |
| Genetic reagent (*D. melanogaster*) | vpoDN-SS1, R31D07-GAL4.AD (attP40); R52F12-GAL4.DBD (attP2) | Bloomington Stock Centre | RRID:BDSC_86868 | |
| Genetic reagent (*D. melanogaster*) | SPSN1, VT058873-GAL4.AD (attP40); VT003280-GAL4.DBD (attP2) | Bloomington Stock Centre | RRID:BDSC_86834 | |
| Genetic reagent (*D. melanogaster*) | SPSN2, VT058873-GAL4.AD (attP40); VT033490-GAL4.DBD (attP2) | Bloomington Stock Centre | RRID:BDSC_86870 | |

*Continued*

| Reagent type (species) or resource | Designation | Source or reference | Identifiers | Additional information |
|---|---|---|---|---|
| Genetic reagent (*D. melanogaster*) | SAG1, VT050405-GAL4.AD (attP40); VT007068-GAL4. DBD (attP2) | Bloomington Stock Centre | RRID:BDSC_66875 | |
| Strain, strain background (*Escherichia coli*) | DH5α | New England Biolabs | RRID:AB_10015282 | For recombinant DNA cloning: |
| Antibody | Anti-HA (rat monoclonal antibody, clone 3F10) | Roche | RRID:AB_390919 | 1:20 |
| Antibody | Anti-GFP (rabbit Polyclonal Antibody) | Molecular Probes | RRID:AB_221570 | 1:100 |
| Antibody | Goat anti-rabbit Alexa Fluor 488 (goat polyclonal antibody) | Molecular Probes | RRID:AB_143165 | 1:250 |
| Antibody | Goat anti-rabbit Alexa Fluor 546 (goat polyclonal antibody) | Molecular Probes | RRID:AB_2534077 | 1:250 |
| Antibody | Goat anti-rabbit Alexa Fluor 647 (goat polyclonal antibody) | Molecular Probes | RRID:AB_2535813 | 1:250 |
| Antibody | Goat anti-rat Alexa Fluor 647 (goat polyclonal antibody) | Molecular Probes | RRID:AB_141778 | 1:250 |
| Sequence-based reagent | pUAST-GGTmSP FRTGFPstopFRT gBlock (FRT underlined) | IDT | Soller Lab | GAATTGGGAATTCGTTAACAGATCTGCGATCG CGGCCCGGGGATCTTGAAGTTCCTATTCCGAAG TTCCTATTCTCTAGAAAGTATAGGAACTTCAGA GCGCTTTTGAAGCTAGCTAAAGAGCCTGCTAA AGCAAAAAAGAAGTCACCATGGTGTCGAGCG CAAGCAAGGGCGAGGAGCTGTTCACCGGGGT GGTGCCCATCCTGGTCGAGCTGGACGGCGAC GTAAACGGCCACAAGTTCAGCGTGTCCGGCGA GGGCGAGGGCGATGCCACCTACGGCAAGCTG ACCCTGAAGTTCATCTGCACCACCGGCAAGCT GCCCGTGCCCTGGCCCACCCTCGTGACCACC CTGACCTACGGCGTGCAGTGCTTCAGCCGCTA CCCCGACCACATGAAGCAGCACGACTTCTTCA AGTCCGCCATGCCCGAAGGCTACGTCCAGGAG CGCACCATCTTCTTCAAGGACGACGGCAACTA CAAGACCCGCGCCGAGGTGAAGTTCGAGGGC GACACCCTGGTGAACCGCATCGAGCTGAAGGG CATCGACTTCAAGGAGGACGGCAACATCCTGG GGCACAAGCTGGAGTACAACTACAACAGCCAC AACGTCTATATCATGGCCGACAAGCAGAAGAAC GGCATCAAGGTGAACTTCAAGATCCGCCACAAC ATCGAGGACGGCAGCGTGCAGCTCGCCGACCA CTACCAGCAGAACACCCCCATCGGCGACGGCC CCGTGCTGCTGCCCGACAACCACTACCTGAGC ACCCAGTCCGCCCTGAGCAAAGACCCCAACGA GAAGCGCGATCACATGGTCCTGCTGGAGTTCG TGACCGCCGCCGGGATCACTCTCGGCATGGAC GAGCTGTACAAGTACTCAGATCTTTGCAAGCTT GTAGAGTTTCCCATTTAATAATTCATATTATCTCGA ATCTAGTCAATTACGGCTTTCCTCAAATAGAAAAA TAAAAAAATGAAAAAATGCACTTGCCATTTAAAC TTAGACGCGATAACGAATTCCGGGGGATCTTGAAGT TCCTATTCCGAAGTTCCTATTCTCTAGAAAGTATAGGAA CTTCAGAGCGCTTTTGAAGCTGCGGCCGCGGCTCG ACGGTATCGATAAGCTTG |
| Software, algorithm | GraphPad Prism | GraphPad Prism | RRID:SCR_002798 | Software |
| Software, algorithm | Fiji | Fiji | RRID:SCR_002285 | Software |

## Fly strains and husbandry

Flies were kept on standard cornmeal-agar food (1% industrial-grade agar, 2.1% dried yeast, 8.6% dextrose, 9.7% cornmeal, and 0.25% Nipagin, all in [w/v]) in a 12-hour light: 12-hour dark cycle. Propionic acid was omitted from fly food as acidity affects egg laying (*Gou et al., 2014*). Genetic crosses

were done in vials and kept at low density to ensure larvae were not competing for food and if necessary, additional live yeast was added. For all behavioural assays, virgin and mated Canton-S were used as controls. Virgin females, for example, from crosses of *GAL4* with *UASmSP*, were collected after emergence within a 5-hour window and well-fed with live yeast sprinkled on food for maximum egg production and allowed to sexually mature (3–5 days).

To recombine second chromosome inserts for *split-GAL4AD* (*attP40*) and third chromosome *split-GAL4DBD* (*attP2*), standard genetic crossing schemes were used and final stocks were balanced with CyO and TM3 Sb (combined from ST and CT stock, see Key Resources Table). *Split-Gal4AD* and *DBD* combination lines were then crossed to *UASmSP*. For meiotic recombination, final stocks were validated by behavioural analysis for *UAS mSP*, for *flp* with *eFeG UASCD8GFP* to monitor GFP expression and for *otdflp UASstopTrpA* and *otdflp UASstopTNT* by crossing to *elavGAL4* and monitored by lethality.

For enhanced recombination with *flp*, virgin females were transferred to 30°C after eclosion and kept for 5 days at this temperature before performing the behavioural assays. For induction of neuronal activity by temperature-sensitive TrpA1, females were kept at 30°C.

To make *UAS FRTstopFRT mSP*, a gBlock (IDT) stop cassette with the FRT sequences used in the eFeG plasmid (*Haussmann et al., 2008*) was inserted into NotI cut *pUAST-GGTmSP* (gift from T. Aigaki) by Gibson assembly. In the stop-cassette, the *FRT* sequence is followed by a *GFP* with a 3'UTR from *ewg* containing polyA site 1 from intron 6 (*Haussmann et al., 2011*). Flies were transformed by P-element-mediated transgenesis and inserts on each chromosome were established that show a robust PMR with *dsxflp* indistinguishable from mated females.

## Behavioural analysis

Females were examined for the main post-mating behaviours receptivity and oviposition as described previously and as follows (*Soller et al., 1999*; *Soller et al., 2006*). To generate mated females, one female and three males were added to fly vials and observed until mating and males were removed after mating. For receptivity tests, mature 3–7-day-old virgin or mated females were added to fly vials (95 mm length and 24 mm diameter) containing Canton S males with an aspirator and observed for 1 hour, generally three females and seven males. For these experiments, males were separated from females at least 1 day before the experiment. Receptivity tests were done in the afternoon with virgins, or 5–24 hours after mating for controls. For oviposition, females were placed individually in fly vials in the afternoon and the number of eggs laid was counted the next day. Receptivity and oviposition tests were tested were done blinded.

## Statistical analysis

Sample size was based on previous studies, non-blinded and not predetermined by statistical methods (*Soller et al., 1997*; *Haussmann et al., 2013*; *Nallasivan et al., 2021*). Behavioural data are representatives of at least three replicates that were performed on three different days. Statistical analysis of behavioural experiments was performed using GraphPad Prism 9 (GraphPad by Dotmatics, RRID:SCR_002798) using one-way ANOVA followed by pairwise comparisons with Tukey's test.

## Immunohistochemistry and imaging

For the analysis of adult neuronal projection from *UAS CD8GFP, UAS H2BYFP,UASmyrGFP, lexAo-pNLStomato,* or *QUAS mtdtomato3xHA* expressing brains, VNCs or genital tracts, tissues were dissected in PBS (137 mM NaCl, 10 mM phosphate, 2.7 mM KCl, pH 7.4), fixed in 4% (w/v in PBS) paraformaldehyde for 15 minutes, washed three times in PBST (PBS with 1% BSA and 0.3% Triton-X100), then once in PBS for 10 minutes, mounted in Vectashield (Vector Labs) and visualised with confocal microscopy using a Leica TCS SP8. If signals were weak, antibody in-situ stainings were done as described previously (*Haussmann et al., 2008*) for validation using rat anti-HA (MAb 3F10, 1:20; Roche), rabbit anti-GFP (Molecular Probes, 1:100) and visualised with Alexa Fluor 488 (1:250; Molecular Probes or Invitrogen), Alexa Fluor 546 (1:250; Molecular Probes or Invitrogen), or Alexa Fluor 647 (1:250; Molecular Probes or Invitrogen). For imaging, tissues were mounted in Vectashield (Vector Labs).

## Confocal microscopy and image processing

Adult tissues were scanned using a Leica SP8 confocal microscope equipped with a set of fluorescent filters and hybrid detector (HyD). Adult brains were scanned using a 40× HC PL APO 40×/1.30 lens with oil, 1024 × 1024 resolution and 0.96 μm Z-step. VNC and genital tracts were scanned using a HC PL APO CS2 20×/0.75 with oil, 1024 × 1024 resolution and 0.96 μm Z-step. Images were obtained using Leica Application Suite X (LAS X) imaging acquisition software. Raw data files were in LIF format and were processed using FIJI RRID:SCR_002285.

For high-resolution mapping, neurons were identified in the virtual fly brain based on registered *GAL4* expression and traces retrieved for modelling (*Scheffer et al., 2020*; *Phelps et al., 2021*; *Galili et al., 2022*).

## Acknowledgements

We thank T Aigaki, G Barnea, P Soba, WJ Joiner, B Dickson, S Goodwin, C Rezaval, D Anderson, JJ Hodge, A Hidalgo, S Collier, O Raibinina, the Bloomington Stock Centre, the Vienna Drosophila RNAi Center for flies, T Aigaki and WJ Joiner for plasmids, the University of Cambridge Department of Genetics Fly Facility and FlyORF for injections, D Scocchia for help with PCR, and IU Haussmann, YJ Kim, JC Billeter, and J-R Martin for comments on the manuscript. We acknowledge funding by the Biotechnology and Biological Science Research Council (BB/N021827/1 and BB/Y006364/1) to MS.

## Additional information

### Funding

| Funder | Grant reference number | Author |
| --- | --- | --- |
| Biotechnology and Biological Sciences Research Council | BB/Y006364/1 | Matthias Soller |
| Biotechnology and Biological Sciences Research Council | BB/N021827/1 | Matthias Soller |

The funders had no role in study design, data collection and interpretation, or the decision to submit the work for publication.

### Author contributions

Mohanakarthik P Nallasivan, Conceptualization, Resources, Formal analysis, Investigation, Visualization, Methodology, Performed genetic experiments and imaging; Deepanshu ND Singh, Resources, Data curation, Formal analysis, Investigation, Performed genetic experiments and imaging; Mohammed Syahir RS Sahir, Formal analysis, Investigation, Performed genetic experiments; Matthias Soller, Conceptualization, Resources, Formal analysis, Supervision, Funding acquisition, Validation, Investigation, Visualization, Methodology, Writing – original draft, Project administration, Writing – review and editing, Designed and performed genetic experiments and analyzed data, Wrote the manuscript with support from MPN

### Author ORCIDs

Deepanshu ND Singh ![ORCID] https://orcid.org/0000-0003-3912-349X
Matthias Soller ![ORCID] https://orcid.org/0000-0003-3844-0258

### Ethics

Ethical review and approval was not required for this study because this study was conducted with an invertebrate model – fruit flies (Drosophila melanogaster). Experiments with invertebrates are not regulated by law.

Reviewer #1 (Public review): https://doi.org/10.7554/eLife.98283.3.sa1
Reviewer #2 (Public review): https://doi.org/10.7554/eLife.98283.3.sa2

Reviewer #3 (Public review): https://doi.org/10.7554/eLife.98283.3.sa3
Author response https://doi.org/10.7554/eLife.98283.3.sa4

## Additional files

### Supplementary files
MDAR checklist

### Data availability
Brain and VNC images for splitGal4 combinations of SP Response Inducing Neurons have been deposited in Virtual Fly Brain and will be published under the following accession numbers: VFB_ x0000000-9. All data generated or analysed during this study are included in the paper and supplementary files; source data files are provided for all figures.

The following dataset was generated:

| Author(s) | Year | Dataset title | Dataset URL | Database and Identifier |
|---|---|---|---|---|
| Nallasivan MP, Singh DND, Saleh MSRS, Soller M | 2025 | Sex Peptide Response Inducing Neurons (SPRINz) | https://virtualflybrain. org/reports/ Nallasivan2026 | Virtual Fly Brain, VFB_ x0000000-9 |

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
