## [Editor Report · eLife Assessment]

This study delivers **valuable** new insights into the neural circuits involved in post-mating responses (PMR) in *Drosophila* females, supported by **convincing** evidence that the circuits for mating receptivity and egg laying are distinct. The new experimental evidence adds to the current understanding of the neural circuits and molecular mechanisms underpinning PMR.

---

## [Referee Report · Reviewer #1 (Public review)]

Summary:

Authors explore how sex-peptide (SP) affects post-mating behaviours in adult females, such as receptivity and egg laying. This study identifies different neurons in the adult brain and the VNC that become activated by SP, largely by using an intersectional gene expression approach (split-GAL4) to narrow down the specific neurons involved. They confirm that SP binds to the well-known Sex Peptide Receptor (SPR), initiating a cascade of physiological and behavioural changes related to receptivity and egg laying.

Comments on revised version:

The authors have substantially strengthened the manuscript in response to our main concerns.

In particular, they now explicitly test multiple established PMR nodes (including SAG/SPSN as well as pC1, OviDN/OviEN/OviIN and vpoDN), which helps separate direct SP targets from downstream PMR circuitry and supports their interpretation that some of these known nodes can affect receptivity without necessarily inducing oviposition. They also addressed key technical/clarity points: the requested head/trunk expression controls are provided (Suppl Fig S1), and the VT003280 annotation is corrected (now FD6 rather than "SAG driver"). Overall, these additions make the central conclusion, that distinct CNS neuron subsets ("SPRINz") are sufficient to elicit PMR components, more convincing, and the added comparisons with genital tract expressing lines further argue against a simple "periphery only" explanation.

---

## [Referee Report · Reviewer #2 (Public review)]

Sex peptide (SP) transferred during mating from male to female induces various physiological responses in the receiving female. Among those, the increase in oviposition and decrease in sexual receptivity are very remarkable. Naturally, a long standing and significant question is the identify of the underlying sex peptide target neurons that express the SP receptor and are underlying these responses. Identification of these neurons will eventually lead to the identification of the underlying neuronal circuitry.

The Soller lab has addressed this important question already several years ago (Haussmann et al. 2013), using relevant GAL4-lines and membrane-tethered SP. The results already showed that the action of SP on receptivity and oviposition is mediated by different neuronal subsets and hence can be separated. The GAL4-lines used at that time were, however, broad, and the individual identity of the relevant neurons remained unclear.

In the present paper, Nallasivan and colleagues carried this analysis a significant step further, using new intersectional approaches and transsynaptic tracing.

Strength:

The intersectional approach is appropriate and state-of-the art. The analysis is a very comprehensive tour-de-force and experiments are carefully performed to a high standard. The authors also produced a useful new transgenic line (UAS-FRTstopFRT mSP). The finding that neurons in the brain (head) mediate the SP effect on receptivity, while neurons in the abdomen and thorax (ventral nerve cord or peripheral neurons) mediate the SP effect on oviposition, is a significant step forward in the endavour to identify the underlying neuronal networks and hence a mechanistic understanding of SP action. The analysis identifies a small set of neurons underlying SP responses. Some are part of the post-mating circuitry aind influence receptivity, while other are likely involved in higher order sensory processing. Though these results are not entirely unexpected, they are novel and represent a significant step forwards as the analysis is at a much higher resolution as previous work.

Weakness:

Though the analysis is at a much higher resolution as previous work on SP targets, it does not yet reach the resolution of single neuronal cell types. The last paragraph in the discussion rightfully speculates about the neurochemical identity of some of the intersection neurons (e.g. dopaminergic P1 neurons, NPF neurons). These suggested identities could have been confirmed by straight-forward immunostainings agains NPF or TH, for which antisera are available. Moreover, specific GAL4 lines for NPF or P1 or at least TH neurons are available which could be used to express mSP to test whether SP activation of those neurons is sufficient to trigger the SP effect. Moreover, the conclusion that SP target neurons operate as key integrators of sensory information for decision of behavioural outputs needs further experimental confirmation.

---

## [Referee Report · Reviewer #3 (Public review)]

Summary:

This paper reports new findings regarding neuronal circuitries responsible for female post-mating responses (PMRs) in Drosophila. The PMRs are induced by sex peptide (SP) transferred from males during mating. The authors sought to identify SP target neurons using a membrane-tethered SP (mSP) and a collection of GAL4 lines, each containing a fragment derived from the regulatory regions of the SPR, fru, and dsx genes involved in PMR. They identified several lines that induced PMR upon expression of mSP. Using split-GAL4 lines, they identified distinct SP-sensing neurons in the central brain and ventral nerve cord. Analyses of pre- and post-synaptic connection using retro- and trans-Tango placed SP target neurons at the interface of sensory processing interneurons that connect to two common post-synaptic processing neuronal populations in the brain. The authors proposed that SP interferes with the processing of sensory inputs from multiple modalities.

Strengths:

Besides the main results described in the summary above, the authors discovered the following:

(1) Reduction of receptivity and induction of egg-laying are separable by restricting the expression of membrane-tethered SP (mSP): head-specific expression of mSP induces reduction of receptivity only, whereas trunk-specific expression of mSP induces oviposition only. Also, they identified a GAL4 line (SPR12) that induced egg laying but did not reduce receptivity.

(2) Expression of mSP in the genital tract sensory neurons does not induce PMR. The authors identified three GAL4 drivers (SPR3, SPR 21, and fru9), which robustly expressed mSP in genital tract sensory neurons but did not induce PMRs. Also, SPR12 does not express in genital tract neurons but induces egg laying by expressing mSP.

---

## [Author Response]

**Public Reviews:**

**Reviewer #1 (Public Review):**
Areas of improvement and suggestions:(1) "These results suggest the SP targets interneurons in the brain that feed into higher processing centers from different entry points likely representing different sensory input" and "All together, these data suggest that the abdominal ganglion harbors several distinct type of neurons involved in directing PMRs"The characterization of the post-mating circuitry has been largely described by the group of Barry Dickson and other labs. I suggest ruling out a potential effect of mSP in any of the well-known post-mating neuronal circuitry, i.e: SPSN, SAG, pC1, vpoDN or OviDNs neurons. A combination of available split-Gal4 should be sufficient to prove this.

We agree that this information is important to distinguish neurons which are direct SP targets from neurons which are involved in directing reproductive behaviors. We have now tested drivers for these neurons and added these data in Fig 3 (SAG neurons) and as Suppl Figs S4 (SPSN and genital tract neuron drivers SPR3 and SPR21), Suppl Fig S6 (overlap in single cell expression atlas), Suppl Fig S7 (overlap of SPSN split drivers with SPR8, fru11/12 and dsx split drivers in the brain inducing PMRs) and Suppl Fig S9 (pC1, OviDNs, OviENs, OviINs and vpoDN).

The newly added data are in full support of our conclusion that SP targets central nervous system neurons, which we termed SP Response Inducing Neurons (SPRINz). In particular, we find lines that express in genital tract neurons, but do not induce an SP response (Supp Figs S4, S7 and S10) or do not express in genital tract neurons and induce an SP response (Fig 2 and Supp Fig S2).

We have analysed the expression of SPSN in the brain and VNC and find expression in few neurons (Suppl Fig S4). This result is consistent with expression of the genes driving SPSN expression in the single cell expression atlas indicating overlap of expression in very few neurons (Suppl Fig S6). We have already shown that FD6 (VT003280) which is part of the SPSN splitGal4 driver, expresses in the brain and VNC and can induce PMRs from SP expression (Fig 4).

We have taken this further to test another SPSN driver (VT058873) in combination with SPR8, fru11/12 and dsx and find PMRs induced by mSP expression (Suppl Fig S7). Moreover, if we restrict expression of mSP to the brain with otdflp we can induce PMRs from mSP expression and obtain the same response by activating these brain neurons (Suppl Fig S7). We note that the VT058873 ∩ fru11/12 intersection in combination with otdflp stopmSP or stopTrpA1 in the head, did not result in PMRs. Here, PMR inducing neurons likely reside in the VNC, but currently no tools are available to test this further.

We further tested pC1, OviDNs, OviENs, OviINs and vpoDN for induction of PMRs from expression of mSP. We are pleased to see that OviEN-SS2s, OviIN-SS1 and vpoDN splitGAl4 drivers can reduce receptivity, but not induce oviposition (Suppl Fig S8). We predicted such drivers based on previously published data (Haussmann et al. 2013), which we now validated.

(2) Authors must show how specific is their "head" (elav/otd-flp) and "trunk" (elav/tsh) expression of mSP by showing images of the same constructs driving GFP.

The expression pattern for tshGAL, which expresses in the trunk is already published (Soller et al., 2006). We have added images for “head” expression for tshGAL and adjusted our statement to be pre-dominantly expressed in the VNC in Suppl Fig 1.

(3) VT3280 is termed as a SAG driver. However, VT3280 is a SPSN specific driver (Feng et al., 2014; Jang et al., 2017; Scheunemann et al., 2019; Laturney et al., 2023). The authors should clarify this.

According to the reviewers suggestion, we have clarified the specificity of VT003280 and now say that this is FD6.

(4) Intersectional approaches must rule out the influence of SP on sex-peptide sensing neurons (SPSN) in the ovary by combining their constructs with SPSN-Gal80 construct. In line with this, most of their lines targets the SAG circuit (4I, J and K). Again, here they need to rule out the involvement of SPSN in their receptivity/egg laying phenotypes. Especially because "In the female genital tract, these split-Gal4 combinations show expression in genital tract neurons with innervations running along oviduct and uterine walls (Figures S3A-S3E)".

We agree with this reviewer that we need a higher resolution of expression to only one cell type. However, this is a major task that we will continue in follow up studies.

In principal, use of GAL80 is a valid approach to restrict expression, if levels of GAL80 are higher than those of GAL4, because GAL80 binds GAL4 to inhibit its activity. Hence, if levels of GAL80 are lower, results could be difficult to interpret.

(5) The authors separate head (brain) from trunk (VNC) responses, but they don't narrow down the neural circuits involved on each response. A detailed characterization of the involved circuits especially in the case of the VNC is needed to (a) show that the intersectional approach is indeed labelling distinct subtypes and (b) how these distinct neurons influence oviposition.

Again, we agree with this reviewer that we need a higher resolution of expression to only one cell type. However, this is a major task that we will continue in follow up studies.

**Reviewer #2 (Public Review):**
Strength:The intersectional approach is appropriate and state-of-the art. The analysis is a very comprehensive tour-de-force and experiments are carefully performed to a high standard. The authors also produced a useful new transgenic line (UAS-FRTstopFRT mSP). The finding that neurons in the brain (head) mediate the SP effect on receptivity, while neurons in the abdomen and thorax (ventral nerve cord or peripheral neurons) mediate the SP effect on oviposition, is a significant step forward in the endavour to identify the underlying neuronal networks and hence a mechanistic understanding of SP action. Though this result is not entirely unexpected, it is novel as it was not shown before.

We thank reviewer 2 for recognizing the advance of our work.

Weakness:Though the analysis identifies a small set of neurons underlying SP responses, it does not go the last step to individually identify at least a few of them. The last paragraph in the discussion rightfully speculates about the neurochemical identity of some of the intersection neurons (e.g. dopaminergic P1 neurons, NPF neurons). At least these suggested identities could have been confirmed by straight-forward immunostainings agains NPF or TH, for which antisera are available. Moreover, specific GAL4 lines for NPF or P1 or at least TH neurons are available which could be used to express mSP to test whether SP activation of those neurons is sufficient to trigger the SP effect.

We appreciate this reviewers recognition of our previous work showing that receptivity and oviposition are separable. As pointed out we have now gone one step further and identified in a tour de force approach subsets of neurons in the brain and VNC.

We agree with this reviewer that we need a higher resolution of expression to only one cell type. As pointed out by this reviewer, the neurochemical identity is an excellent suggestions and will help to further restrict expression to just one type of neuron. However, this is a major task that we will continue in follow up studies.

**Reviewer #3 (Public Review):**
Strengths:Besides the main results described in the summary above, the authors discovered the following:(1) Reduction of receptivity and induction of egg-laying are separable by restricting the expression of membrane-tethered SP (mSP): head-specific expression of mSP induces reduction of receptivity only, whereas trunk-specific expression of mSP induces oviposition only. Also, they identified a GAL4 line (SPR12) that induced egg laying but did not reduce receptivity.(2) Expression of mSP in the genital tract sensory neurons does not induce PMR. The authors identified three GAL4 drivers (SPR3, SPR 21, and fru9), which robustly expressed mSP in genital tract sensory neurons but did not induce PMRs. Also, SPR12 does not express in genital tract neurons but induces egg laying by expressing mSP.

We thank reviewer 2 for recognizing these two important points regarding the SP response that point to a revised model for how the underlying circuitry induces the post-mating response. To further substantiate these findings we now have added a splitGal4 nSyb ∩ ppk which expresses in genital tract neurons, but does not induce PMRs from mSP expression.

Weaknesses:(1) Intersectional expression involving ppk-GAL4-DBD was negative in all GAL4AD lines (Supp. Fig.S5). As the authors mentioned, neurons may not intersect with SPR, fru, dsx, and FD6 neurons in inducing PMRs by mSP. However, since there was no PMR induction and no GAL4 expression at all in any combination with GAL4-AD lines used in this study, I would like to have a positive control, where intersectional expression of mSP in ppk-GAL4-DBD and other GAL4-AD lines (e.g., ppk-GAL4-AD) would induce PMR.

We have added a positive control for ppk expression by combining the ppk-DBD line with a nSyb-AD which expresses in all neurons in Supp Fig S8. This experiment confirms our previous observations that ppk splitGal4 in combination with other drivers does not induce an SP response despite driving expression in genital tract neurons. We have expanded the discussion section to point out that we have identified additional cells in the brain expressing ppkGAL4, but expression of split-GAL4 ppk is absent in these cells. Part of this work has previously been published (Nallasivan et al. 2021). Accordingly, we amended the text to say when expression was achieved with ppkGAL or ppk splitGAL4.

(2) The results of SPR RNAi knock-down experiments are inconclusive (Figure 5). SPR RNAi cancelled the PMR in dsx ∩ fru11/12 and partially in SPR8 ∩ fru 11/12 neurons. SPR RNAi in dsx ∩ SPR8 neurons turned virgin females unreceptive; it is unclear whether SPR mediates the phenotype in SPR8 ∩ fru 11/12 and dsx ∩ SPR8 neurons.

We agree with this reviewer that the interpretation of the SPR RNAi results are complicated by the fact that SP has additional receptors (Haussmann et al 2013). The results are conclusive for all three intersections when expressing UAS mSP in SPR RNAi with respect to oviposition, e.g. egg laying is not induced in the absence of SPR. For receptivity, the results are conclusive for dsx ∩ fru11/12 and partially for SPR8 ∩ fru 11/12.

Potentially, SPR RNAi knock-down does not sufficiently reduce SPR levels to completely reduce receptivity in some intersection patterns, likely also because splitGal4 expression is less efficient.

Why SPR RNAi in dsx ∩ SPR8 neurons turned virgin females unreceptive is unclear, but we anticipate that we need a higher resolution of expression to only one cell type to resolve this unexpected result. However, this is a major task that we will continue in follow up studies.

SPR RNAi knock-down experiments may also help clarify whether mSP worked autocrine or juxtacrine to induce PMR. mSP may produce juxtacrine signaling, which is cell non-autonomous.

Whether membrane-tethered SP induces the response in a autocrine manner is an import aspect in the interpretation of the results from mSP expression.

Removing SPR by SPR RNAi and expression of mSP in the same neurons did not induce egg laying for all three intersection and did not reduce receptivity for dsx ∩ fru11/12 and for SPR8 ∩ fru 11/12. Accordingly, we can conclude that for these neurons the response is induced in an autocrine manner.

We have added this aspect to the discussion section.